# Toward an integrated map of genetic interactions in cancer cells

Benedikt Rauscher[1,2], Florian Heigwer[1,2] (iD), Luisa Henkel[1,2], Thomas Hielscher[3], Oksana Voloshanenko[1,2] & Michael Boutros[1,2,*] (iD)

## Abstract

Cancer genomes often harbor hundreds of molecular aberrations. Such genetic variants can be drivers or passengers of tumorigenesis and create vulnerabilities for potential therapeutic exploitation. To identify genotype-dependent vulnerabilities, forward genetic screens in different genetic backgrounds have been conducted. We devised MINGLE, a computational framework to integrate CRISPR/Cas9 screens originating from different libraries building on approaches pioneered for genetic network discovery in model organisms. We applied this method to integrate and analyze data from 85 CRISPR/Cas9 screens in human cancer cells combining functional data with information on genetic variants to explore more than 2.1 million gene-background relationships. In addition to known dependencies, we identified new genotype-specific vulnerabilities of cancer cells. Experimental validation of predicted vulnerabilities identified GANAB and PRKCSH as new positive regulators of Wnt/β-catenin signaling. By clustering genes with similar genetic interaction profiles, we drew the largest genetic network in cancer cells to date. Our scalable approach highlights how diverse genetic screens can be integrated to systematically build informative maps of genetic interactions in cancer, which can grow dynamically as more data are included.

**Keywords** cancer; epistasis; genetic interactions; networks; synthetic lethality
**Subject Categories** Cancer; Computational Biology; Genome-Scale & Integrative Biology
**Mol Syst Biol. (2018) 14: e7656**

## Introduction

Genes rarely function in isolation to affect phenotypes at the cellular or organismal level. Many studies have described how genes act in complex networks to maintain homeostasis by fine-tuning cellular or organismal reactions to internal or external stimuli (Bergman & Siegal, 2003). A loss of genetic buffering can result in the emergence of diseases such as cancer (Hartwell *et al*, 1997; Hartman *et al*, 2001). In turn, mutations can create genetic vulnerabilities in cancer cells, for example, by deactivating one of two genetically buffered pathways (Luo *et al*, 2009; Torti & Trusolino, 2011; Nagel *et al*, 2016). Therapeutic approaches attempt to exploit such events by selectively inducing cell death in cancer cells while causing little harm to normal cells (Kaelin, 2005; Nijman, 2011).

To systematically identify genetic interactions, pairwise gene knockout or knockdown experiments can be performed (Mani *et al*, 2008). In cases where a measured fitness defect of the double mutant is stronger than expected based on the two single mutant phenotypes, the interaction is called aggravating or synthetic lethal (Bridges, 1922). In contrast, a buffering (or alleviating) interaction is observed when the double mutant's measured phenotype is weaker than expected. Arrayed screens performed by mating of loss-of-function mutant yeast strains have pioneered combinatorial screening (Tong *et al*, 2001; Davierwala *et al*, 2005; Baryshnikova *et al*, 2010; Costanzo *et al*, 2010, 2016). Methods of pairwise gene perturbation were later extended using combinatorial RNA interference (RNAi) to map genetic interactions in cultured metazoan cells (Byrne *et al*, 2007; Horn *et al*, 2011; Laufer *et al*, 2013; Snijder *et al*, 2013; Fischer *et al*, 2015; Srivas *et al*, 2016). However, screening of all pairwise gene combinations scales poorly with increasing genome size and novel approaches are necessary to facilitate the generation of large genetic interaction maps of complex organisms while minimizing cost and experimental effort.

Genome-scale perturbation screens can now be efficiently performed in many cell lines using CRISPR/Cas9 (Barrangou, 2014; Doudna & Charpentier, 2014; Wang *et al*, 2014; Shalem *et al*, 2015; Heigwer *et al*, 2016; Horlbeck *et al*, 2016) or RNAi (Brummelkamp *et al*, 2002; Sims *et al*, 2011; Kampmann *et al*, 2013) for the targeted perturbation of genes by knockout or knockdown. Since each cell line has a different genetic background, this enables the investigation of genotype-specific vulnerabilities (Garnett *et al*, 2012; Hart *et al*, 2015; Iorio *et al*, 2016; Tzelepis *et al*, 2016; Martin *et al*, 2017; McDonald *et al*, 2017; Steinhart *et al*, 2017; Tsherniak *et al*, 2017; Wang *et al*, 2017). To describe a genetic interaction, previous studies have mostly relied on the definition of "statistical epistasis" introduced by R. A. Fisher (Fisher, 1930). Here, a genetic interaction

1 Division of Signaling and Functional Genomics, German Cancer Research Center (DKFZ), Heidelberg, Germany
2 Department of Cell and Molecular Biology, Medical Faculty Mannheim, Heidelberg University, Heidelberg, Germany
3 Division of Biostatistics, German Cancer Research Center (DKFZ), Heidelberg, Germany
   *Corresponding author. Tel: +49 6221 421950; Fax: +49 6221 421959; E-mail: m.boutros@dkfz.de

is defined as a statistical deviation from the additive combination of two loci in how they affect a phenotype of interest (Phillips, 2008). This definition does not necessarily assume a standardized genetic background and thus provides a theoretical framework applicable to map genetic interactions in cancer cell lines despite the presence of additional confounding mutations. To leverage the community's collective effort to functionally characterize cancer cell lines, it is desirable to combine and analyze genetic screens of different origin in an integrated manner. This, however, is not easily put into practice as various sources of technical variation such as different sgRNA libraries or experimental protocols can affect the data and confound comparative analyses.

Here, we propose a computational framework that integrates CRISPR/Cas9 screens of diverse origin to map genetic interactions in cancer cells. We apply this approach, which we termed MINGLE, to a curated dataset consisting of 85 genome-scale CRISPR/Cas9 screens in 60 different human cancer cell lines generated in various different laboratories (Fig 1A). We first show that a two-step normalization approach can be applied to enable quantitative comparison of phenotypes derived from different screens (Fig EV1A). We then demonstrate how concepts that have previously been applied to map genetic networks in model organisms can be adapted and applied to

this dataset to score gene–gene combinations for genetic interactions. Combining the intrinsic profile of genetic alterations of each cell line present in the dataset with gene-level viability phenotypes, we tested 2.1 million pairwise gene combinations by comparing wild type against altered alleles in cell lines (Fig 1B and C). Using these predictions, we were able to identify new regulators of the Wnt/β-catenin signaling pathway. Our results suggest that the genes *PRKCSH* and *GANAB*, which together form the glucosidase II complex, regulate the secretion of active Wnt ligands. Finally, we functionally clustered genes by the similarity of their interaction profiles and demonstrate that these profiles are informative predictors of functional gene similarity (Fig 1D). We generated a map of genetic interactions in cancer cells by connecting genes with similar profiles and identified network modules with similar functional characteristics.

## Results

### Integrating CRISPR/Cas9 phenotypes from different studies

In order to systematically predict interactions between genes knocked out by CRISPR/Cas9 and genes functionally impaired by

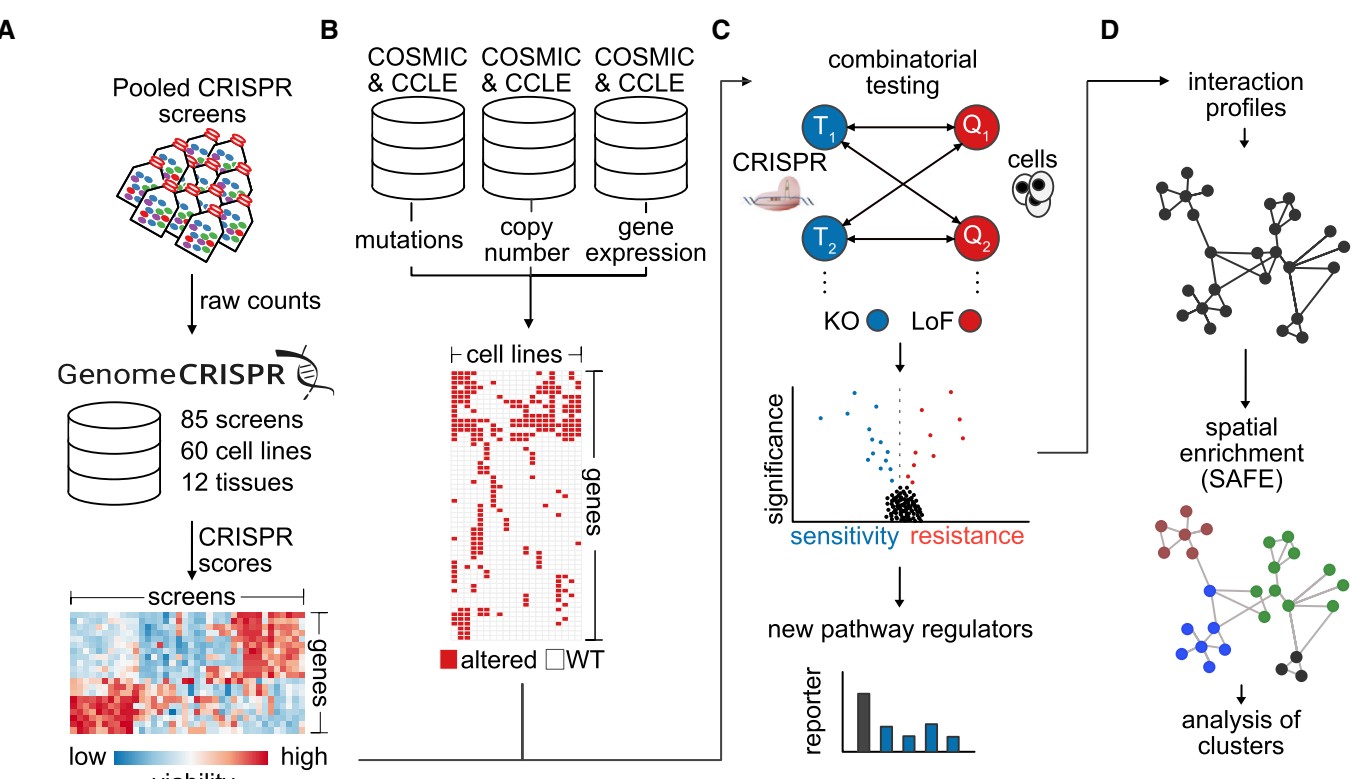

**Figure 1.  An integrated analysis approach to identify genetic interactions in cancer cells.**

A    Data from CRISPR/Cas9 screens in 60 cancer cell lines were re-analyzed and integrated. The results were integrated into a global perturbation response profile.

B    Mutation, copy number, and mRNA expression data from the COSMIC and CCLE databases were combined to create a map of genetic alterations across these cell lines.

C    To identify genetic dependencies between gene combinations that could shed light on the genetic wiring of cancer cells, perturbation response of more than 2.1 million gene–gene combinations was examined to infer genetic interactions.

D    Interaction profiles were calculated for gene combinations based on the correlation of their interactions as determined by interaction scores (π scores). Spatial enrichment analysis was performed to identify functional modules in the network.

mutations in cancer cells, we re-analyzed a set of 85 CRISPR/Cas9 viability screens in 60 cell lines (Fig 1A, Dataset EV1). These screens were performed in different laboratories and vary in terms of library and vector design as well as screening protocols. In order to integrate these data (Fig EV1A), we first calculated gene-level CRISPR scores individually for each screen (average $\log_2$ fold change of sgRNA abundance; Wang *et al*, 2017). As, for example, varying selection times can lead to differences in phenotypic strength, we then quantile-normalized the data to correct for systematic biases between screens. Examination of the resulting dataset revealed considerable batch effects driven primarily by the sgRNA library used for screening (Fig EV1B). These batch effects appeared to be non-systematic differing from gene to gene. For example, cyclin-dependent kinase 7 (*CDK7*) is a gene known to play important roles in both, cell cycle progression and transcription (Fisher, 2005), and is expected to be a broadly essential gene (Hart *et al*, 2017). Accordingly, knockout of *CDK7* consistently led to decreased viability in the majority of experiments. The screens in which no viability phenotype was observed upon *CDK7* knockout were all conducted using the same library (Fig EV1C). Since the cell lines screened with this library are derived from various different tissues and cancer types, a common resistance to *CDK7* knockout seems unlikely. A more probable explanation for the observed batch effect might be the inability of *CDK7* targeting sgRNAs in this library to generate a knockout in the first place. If not considered and corrected, such batch effects can introduce false predictions (Fig EV1D), underlining the requirement of an efficient strategy for their adjustment. To this end, we hypothesized that a gene knockout should, on average, have the same effect across screens, regardless of the library used. We then applied a model-based approach to systematically scan for potential batch effects where the phenotypes generated by one library differed significantly (FDR < 5%) from the observed median phenotype across all libraries. In order to protect real biological effects, we used a robust linear model for testing, which is robust toward strong biological effects present in the data in the form of outliers. In cases, in which a significant difference between the phenotypes generated by one library and the median phenotype across all libraries could be detected, we performed an adjustment by subtracting the estimated difference between the library affected by the batch effect and the remaining libraries (Fig EV1B). It is important to point out, that this approach can be inappropriate when there is a correlation between an sgRNA library and a biological covariate, for example, if most cell lines screened with this specific library are derived from similar tissues. This is not the case for most libraries included in this analysis. For example, the GeCKOv2 and TKOv1 libraries have been used to screen a wide variety of cell lines derived from different tissues and cancer types (Hart *et al*, 2015; Aguirre *et al*, 2016; Steinhart *et al*, 2017). An exception, however, is the screens performed by Wang *et al* (2017) as well as Tzelepis *et al* (2016). In these studies, screens were performed primarily in acute myeloid leukemia (AML) cell lines. In order to preserve such tissue-specific phenotypes through batch correction, our model-based approach allows to include biological covariates such as a cell line's tissue or cancer type into the batch modeling, which can then distinguish between technical and biological variability.

In order to validate our data integration approach, we performed a variety of quality control analyses. First, we clustered all screens based on the normalized CRISPR scores (Figs 2A and EV1F). In many cases, screens that were performed in different laboratories with different libraries but using the same cell line clustered together. Moreover, we observed a tendency for cell lines sharing the same tissue origin to group together. For example, we could identify distinct clusters of AML cell lines and adenocarcinoma cell lines. These results suggest appropriate correction of technical bias, leaving the biological variability across cell lines as the main driver of the clustering. We next assessed whether normalized CRISPR scores can be compared quantitatively across screens. Here, we randomly selected nine core-essential polymerases and plotted normalized CRISPR scores for these genes across screens (Fig 2B). CRISPR scores for essential polymerases were negative and approximately on the same level with no noticeable differences between screens published in different studies, suggesting that quantitative comparison of scores is indeed feasible and that expected negative viability phenotypes of core-essential gene knockouts are preserved throughout normalization. We wondered if the normalization procedure could potentially introduce false phenotypes. Generally, this can be ruled out with the help of non-targeting controls, which, however, were not available for all experiments in our dataset. As a replacement, we therefore selected all screens performed in female cell lines and plotted normalized CRISPR scores for nine randomly selected genes located on the Y chromosome (Fig 2C). We observed CRISPR scores to be approximately 0, implying that no false phenotypes are introduced artificially by the normalization. Next, we determined how well core-essential and non-essential reference genes (Hart *et al*, 2015, 2017) could be separated based on the normalized CRISPR scores by generating precision–recall curves (Fig 2D), based on which we observed good performance across all screens. We further examined if the normalized CRISPR scores could capture well-studied examples of oncogene addiction. Oncogene addiction describes a phenomenon where cancer cells, albeit harboring many molecular aberrations, become strongly dependent on only a single one of them. Reversing this abnormality leads to growth inhibition and apoptosis (Weinstein & Joe, 2006). We selected the well-studied oncogenes *KRAS*, *NRAS*, *BRAF,* and *PIK3CA* and compared the CRISPR scores of cell lines harboring a mutation of these genes to the rest of the cell lines (Fig 2E–H). As expected, we observed considerably stronger phenotypes in the mutated cells as compared to the wild-type cells. Last, we determined whether genetic dependencies previously identified in screens used for our analysis could be reproduced (Fig EV1E). In all cases, we could achieve comparable results to those previously published, corroborating the usage of normalized CRISPR scores for valid interscreen analysis.

## Interactions between gene knockouts and cancer alterations reveal genetic wiring maps

In order to determine genetic interactions, we formed all pairwise combinations between genes knocked out by CRISPR/Cas9 in pooled viability screens (target genes) and genes altered in cancer cells (query genes) (Fig 1C). We only considered genes as queries if they contain an alteration in at least three distinct cell lines (Dataset EV2). A cancer alteration was defined as a somatic mutation, a somatic copy number alteration (SCNA) or differential expression of a gene. We pooled alterations for each gene based on three assumptions: We assumed that (i) a loss of gene copy number behaves similarly to a disruptive somatic mutation (e.g., a frame-shift

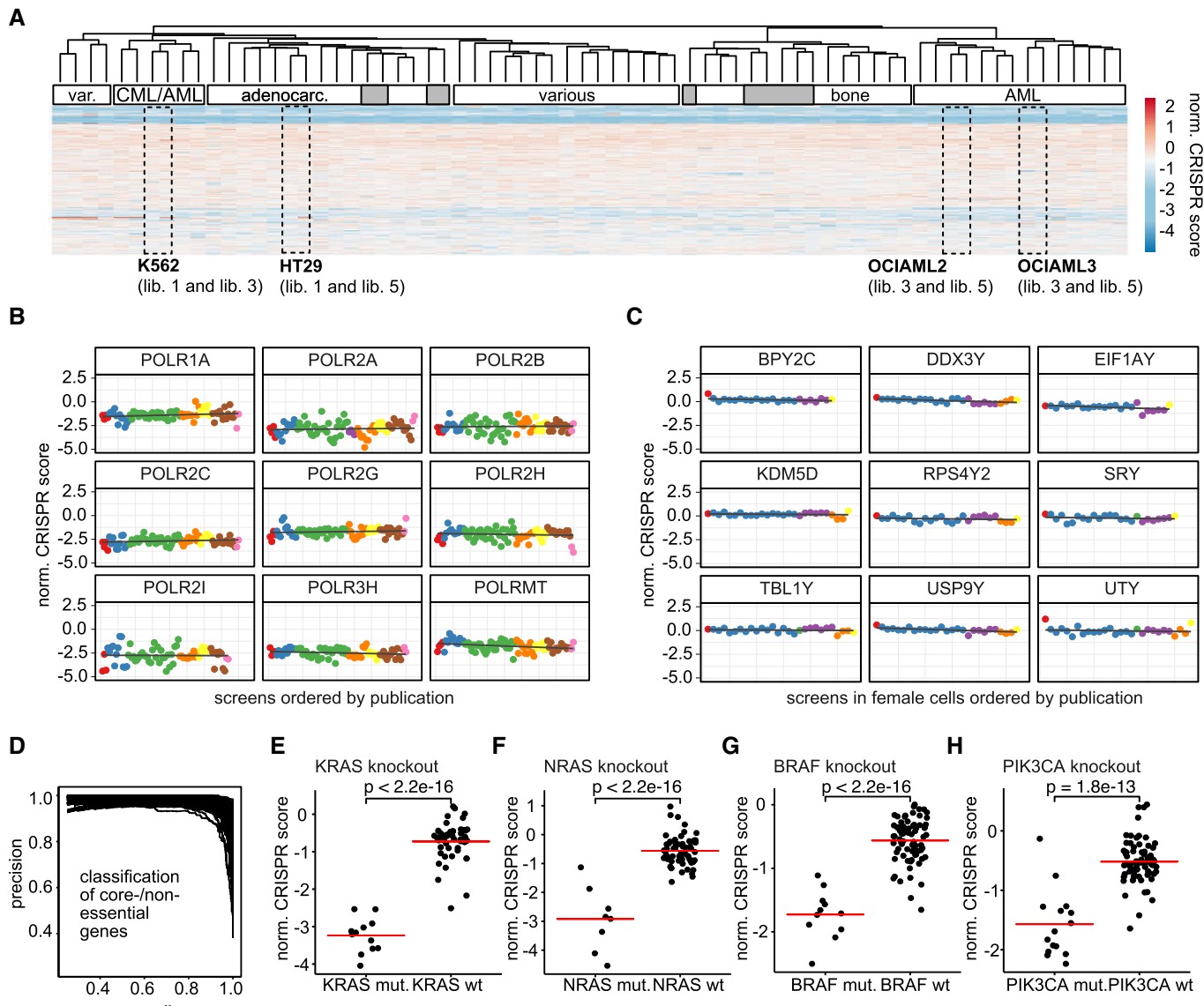

**Figure 2.  Results and quality control of data integration and normalization.**

A    A heat map shows a clustering of normalized CRISPR scores (average log2 fold change of sgRNAs targeting a gene) for genes present in each sgRNA library used in screens included in the analysis. Rectangular windows highlight experiments where screens performed in the same cell line but in different laboratories cluster together. White annotation bars indicate shared biological properties of the cell lines in each cluster. Gray bars indicate the annotated cell line does not fit to the annotation of other cell lines in the same cluster.

B    Normalized CRISPR scores across experiments are displayed for a randomly selected set of nine core-essential polymerases. Each dot corresponds to one screen, and different colors highlight the publications that the data were derived from. More negative CRISPR scores indicate a more negative viability response upon gene knockout.

C    Normalized CRISPR scores across experiments in female cell lines are displayed for a randomly selected set of nine genes located on the Y chromosome serving as non-targeting controls. Colors depict different publications.

D    Precision–recall curves showing the performance of normalized CRISPR scores at distinguishing core-essential from non-essential genes. Each line corresponds to one experiment. High recall while maintaining high precision indicates good performance.

E–H    Comparison of normalized CRISPR scores in a different genetic background for four different control dependencies. Red lines indicate group means. Statistical significance was determined using a two-sided Student's *t*-test. Each data point represents one screening experiment. The groups consist of 12 (KRAS mut.), 44 (KRAS wt), 7 (NRAS mut.), 56 (NRAS wt), 11 (BRAF mut.), 63 (BRAF wt), 15 (PIK3CA mut.) and 59 (PIK3CA wt) data points.

mutation or a nonsense mutation), (ii) a gain of copy number behaves similarly to a gain of gene expression, and that (iii) somatic mutations of the same gene have, on average, a similar functional consequence. Even though these assumptions, especially number 3, do in reality not always hold true, we found them to be a useful

approximation judging by the results we obtained in downstream genetic interaction analyses. In addition, we further refined some of the pooled genetic alterations by manual curation excluding cell lines with alterations known to be functionally dissimilar to other alterations of the same gene. This, however, was only possible for

well-characterized genes. In total, we formed 3.8 million gene pairs of 17,218 target genes and 221 query genes.

Assuming that two genes do in most cases not interact with each other, we first performed a statistical test for each gene pair, comparing normalized CRISPR scores of cells that contain an alteration of the query gene to cells that do not contain the alteration. Here, we used a multilevel model including the cell line corresponding to each data point as a random effect to account for biases that could potentially be introduced when one cell line was screened multiple times. In some cases, we observed high correlation between several query genes (Fig EV2A). This observation can, for example, be explained by a co-deletion of genes that are located close to each other on the genome. For instance, *CDKN2A*, a tumor suppressor gene (Liggett & Sidransky, 1998) located on chromosome band 9p21, is often co-deleted with its surrounding genes (Muller *et al*, 2015). In such cases, it is not possible to determine with which of the two potential query genes a target gene should be predicted to interact. We addressed this by aggregating identical query genes, as determined by the correlation of their model coefficients, into "meta genes" that we then used for downstream analyses (Fig EV2B). To quantify the interaction strength of each gene pair, we calculated π-scores (Fig 3A and B) as described previously (Horn *et al*, 2011; Laufer *et al*, 2013; Fischer *et al*, 2015). Altogether, our analysis predicted 17,545 gene–gene interactions at FDR < 20% (0.8% of total combinations tested after meta gene aggregation).

Examining the proposed interactions, we found that our analysis was able to recover many previously characterized dependencies across several pathways that have been extensively studied in the past (Figs 3 and EV2F–H). For example, we identified many positive interactions (i.e., cells containing an alteration of the query gene are more resistant to perturbation of the target gene) between *TP53* and several genes involved in stabilization of the p53 protein (Fig 3C). In wild-type cells, p53 is kept at low abundance by E3/E4 ubiquitin ligases including, for example, *MDM2* and *MDM4* (Fig EV2G), which can mediate its degradation via the proteasome (Lavin & Gueven, 2006; Frum & Grossman, 2014). Knockout of these ubiquitin ligases likely leads to an accumulation of p53, which might then mediate apoptosis and impede proliferation resulting in a negative viability phenotype. In tumor cells, missense mutations of the *TP53* gene can inhibit p53 degradation (Lavin & Gueven, 2006; Frum & Grossman, 2014) where it can accumulate and act as an oncogene (Oren & Rotter, 2010), which could explain the resistance of *TP53*-mutated cell lines to loss of E2/E3 ubiquitin ligases. An interaction that at first glance might seem surprising is a negative interaction of *TP53* with itself (i.e., cells with a *TP53* mutation are more sensitive to *TP53* knockout). In the context of epistasis, however, this might be explained by the fact that in *TP53* wild-type cells, where *TP53* acts as a tumor suppressor, its knockout leads to a gain of viability phenotype, which is not the case for tumor cells which already harbor mutations in *TP53* (Fig EV2H). Next, we looked at predicted interactions of the *BRAF* oncogene. Unsurprisingly, we found negative interactions with *BRAF* itself as well as *MAP2K1* (MEK1) and *MAPK1* (ERK2), both of which lie downstream of *BRAF* in the MAPK signaling cascade (Seger & Krebs, 1995). In contrast, no interactions were found for upstream components of the pathway such as *KRAS* or *EGFR* (Fig 3D), likely because the constitutive activation of *BRAF* caused by its mutation confers independence on upstream pathway components. Following previous studies (Brockmann *et al*, 2017),

we reasoned that genes that interact specifically with one or few related query genes should be functionally related. We thus selected ten query genes including their predicted interaction partners at FDR < 20% and performed gene set overrepresentation analysis (Kamburov *et al*, 2013) for groups of target genes specifically interacting with one of the selected queries (Fig 3F). Looking at pathways over-represented within the analyzed set of genes, we found several well-characterized relationships linking, for example, mutations of *KRAS*, *NRAS*, or *BRAF* to MAPK signaling, *BCL2* to apoptosis or *TP53* to the stabilization thereof, suggesting a high number of true predictions. In addition, our analysis proposes genetic interactions for many other less well-studied query genes (a full list of predicted interactions can be found in Dataset EV3). To find traits shared between query genes for which high interaction numbers were predicted (Fig EV2E), we performed GO (Ashburner *et al*, 2000) molecular function enrichment analysis (Kuleshov *et al*, 2016). Unsurprisingly, we found that GO terms with the highest enrichment scores were related to transcription factor activity (Fig 3G). Other high-ranking GO terms were related to chromatin remodeling and hormone receptor binding.

We hypothesized that it should be possible to combine functionally related query genes in order to improve prediction of regulators of signaling pathways. Consequently, we combined loss of function mutations of the genes *APC* and *RNF43* (Dataset EV3) into a "Wnt mutation" query metagene. Both, *APC* and *RNF43*, are frequently mutated negative regulators of the Wnt/β-catenin signaling pathway (Polakis, 2012; de Lau *et al*, 2014; Tsukiyama *et al*, 2015; Zhan *et al*, 2017)—a pathway that is aberrantly regulated in various cancers (Polakis, 2012; Giannakis *et al*, 2014; Zhan *et al*, 2017). In the absence of Wnt ligands, APC regulates β-catenin activity *via* the formation of a destruction complex with GSK3β and Axin1, which mediates β-catenin phosphorylation. Phosphorylated β-catenin is then targeted for degradation by the proteasome. Binding of canonical Wnts to Frizzled receptors and LRP5/6 co-receptors on the cell surface inhibits the formation of the destruction complex, which results in β-catenin stabilization and its translocation to the nucleus. Within the nucleus, β-catenin interacts with TCF/LEF transcription factors and activates transcription of Wnt target genes, which mediate cell growth and survival (MacDonald *et al*, 2009). *RNF43* is an E3 ubiquitin ligase that can induce ubiquitination and subsequent degradation of the Wnt–Frizzled complex (MacDonald *et al*, 2009; Clevers & Nusse, 2012), thus inhibiting β-catenin signaling. Consequently, disruptive mutations in *APC* or *RNF43* can promote activation of the pathway. Examining genes predicted to interact with loss-of-function mutations of either *APC* or *RNF43*, we observed many known regulators of Wnt/β-catenin signaling (Fig 3E). Among these, we identified, for example, regulators of Wnt ligand secretion, *TCF7L2* and *CTNNB1* which together form the TCF/β-catenin transcription factor complex, and other genes, which have previously been linked to the Wnt/β-catenin pathway (Chen *et al*, 2014; Ormanns *et al*, 2014).

## Dependency analysis of Wnt pathway alterations reveals novel regulators of Wnt/β-catenin signaling

We hypothesized that among known modulators of Wnt/β-catenin signaling, our analysis should also identify so far uncharacterized pathway regulators. Inactivating mutations of the *RNF43* gene, for

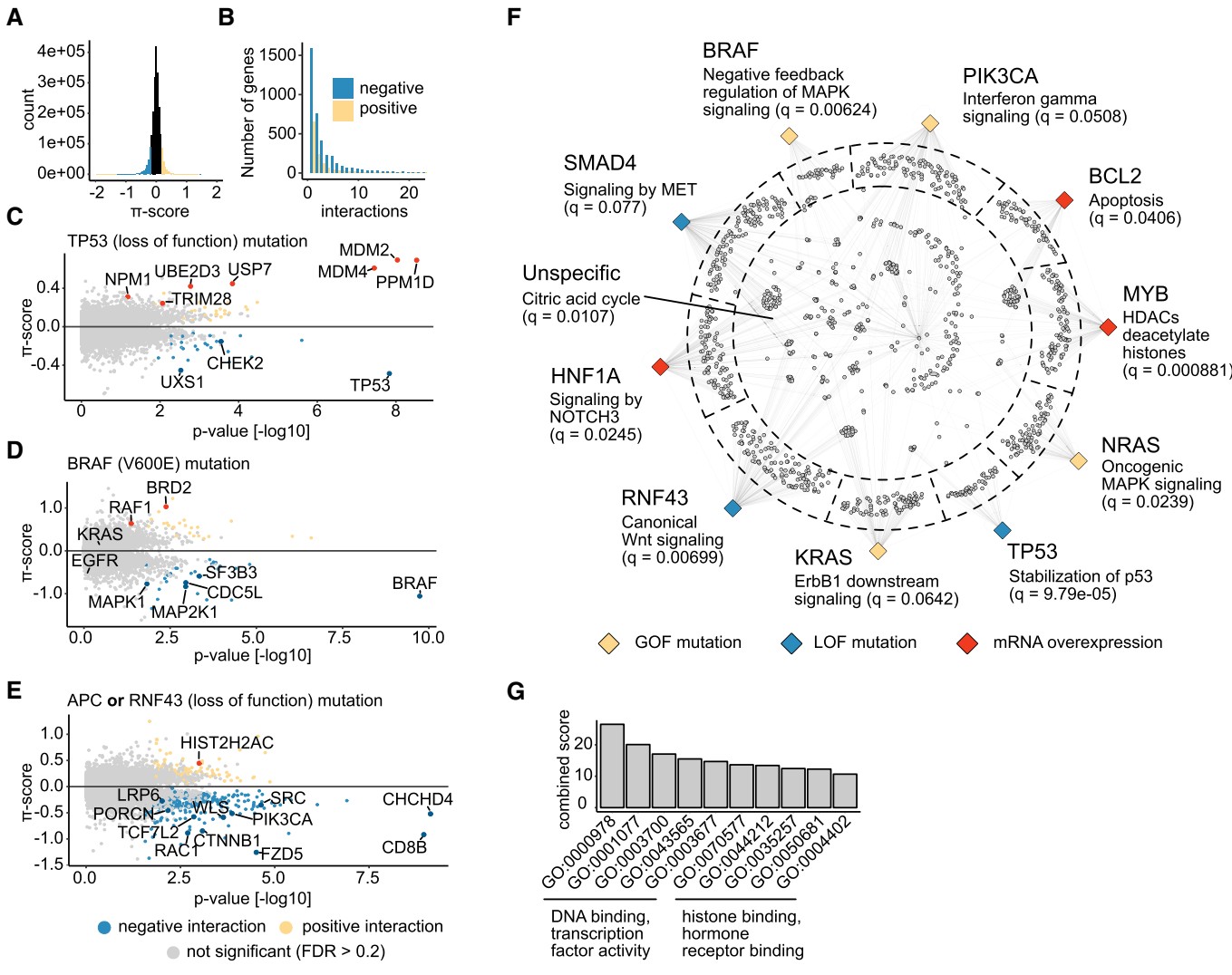

**Figure 3. Results of predicted genetic interactions.**

A    Distribution of π-scores calculated for each pairwise interaction. Negative values indicate negative (aggravating interactions), and positive values indicate positive (buffering) interactions. Values > 0.2 and < −0.2 are colored yellow and blue, respectively.

B    The number of positive and negative interactions per gene. Interactions with a π-score > 0.2 are considered positive, and interactions with a π-score of < −0.2 are considered negative.

C–E    Volcano plots showing genes interacting with TP53 loss-of-function mutations (C), BRAF V600E mutations (D), and APC or RNF43 loss-of-function mutations (E). Each dot corresponds to one gene. Interactions that are significant at FDR < 0.2 are colored in blue in case the interaction is negative or yellow if it is positive. Selected genes are highlighted and labeled.

F    A network graph showing gene set enrichment results for sets of interaction partners. Each of the colored diamonds corresponds to one of 10 selected query alterations. The color of each diamond indicates the type of alteration as described in the legend at the bottom. Each gray dot connected to one or more query gene nodes represents a target gene that interacts (FDR < 0.2) with the query. Gene set enrichment analysis was performed for genes that fall in the same compartment as indicated by the dashed line. Genes in compartments toward the edge interact with one specific query. Genes positioned in the center of the circle have a more promiscuous interaction profile. Selected enriched pathway terms are used to label the query gene nodes.

G    GO terms enriched among 40 query genes with the most interactions ($|\pi| > 0.2$, FDR < 0.2).

example, have previously been shown to confer dependency on Wnt/β-catenin signaling (Jiang *et al*, 2013; Steinhart *et al*, 2017) so we reasoned that negative interactions of *RNF43* could point to positive pathway regulators. Besides known Wnt pathway regulators, our analysis revealed negative interactions between RNF43 and several in this context uncharacterized genes (Dataset EV3). We aimed to experimentally validate these predictions and proceeded by selecting three high-scoring candidate genes reported to be involved

in protein glycosylation (D'Alessio & Dahms, 2015) for follow-up (Fig 4A). Two of these genes, *PRKCSH* and *GANAB*, together form the heterodimeric glucosidase II. The third candidate, *UGP2*, is involved in carbohydrate synthesis (Wang *et al*, 2016). We knocked down each of the candidate genes using at least three different siRNAs (Figs 4B and EV3B, Materials and Methods) or a pool consisting of the same reagents in HEK293T cells (Fig 4B) (Thomas & Smart, 2005). HEK293T cells were chosen as a well-established

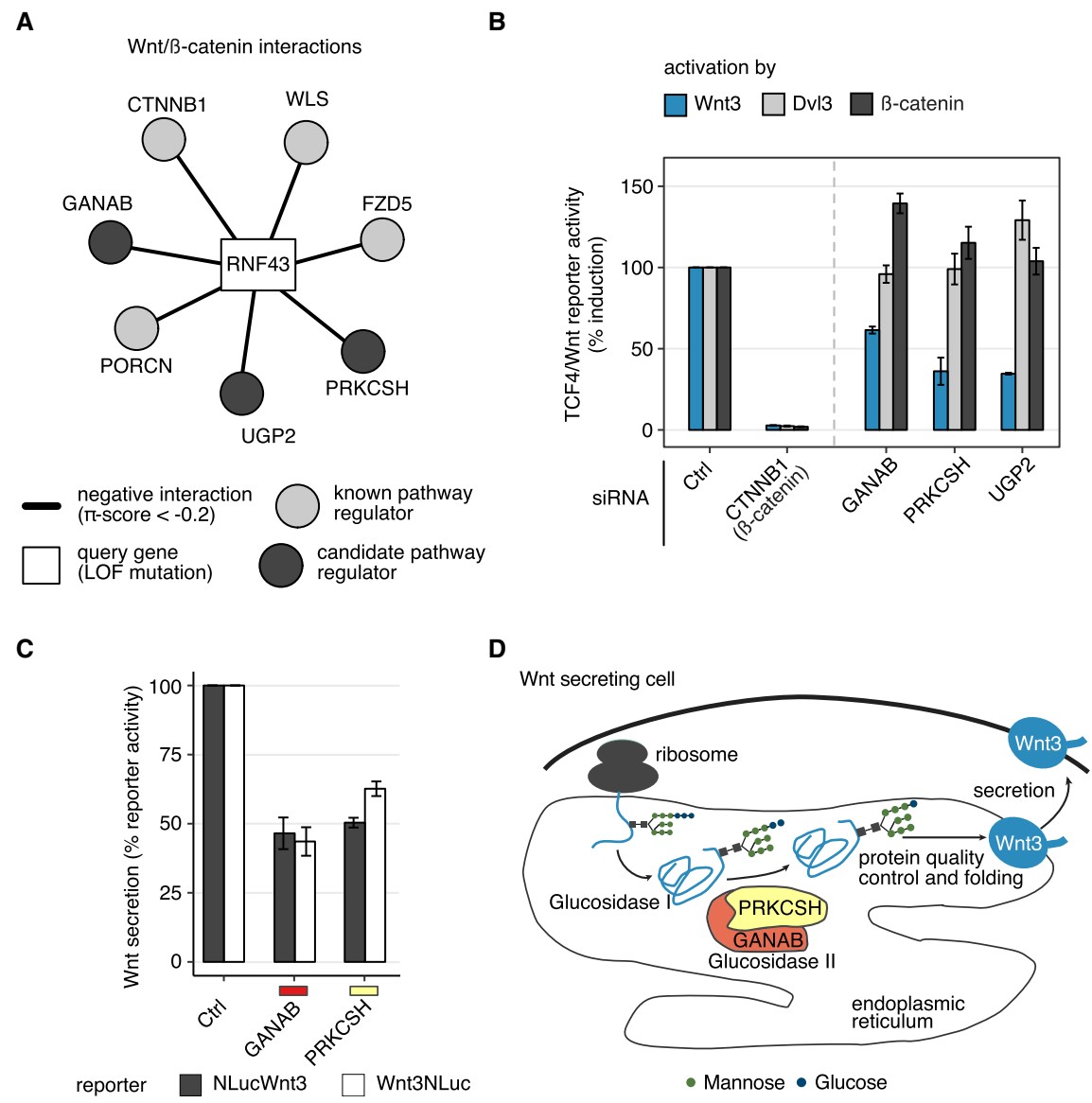

**Figure 4.   Candidate genes *GANAB* and *PRKCSH* regulate Wnt secretion.**

A   Three candidate genes (dark gray circles) interact with the *RNF43* query gene (rectangle), similar to well-characterized pathway components (light gray circles).

B   HEK293T cells were reverse transfected with siRNA pools targeting genes labeled on the *x*-axis. 24 hours after transfection, Wnt signaling was activated by overexpression of Wnt3, Dvl3, or β-catenin plasmids. The TCF4/Wnt Firefly luciferase signal was normalized to the actin-*Renilla* signal. Results are shown as averages of 3–4 independent experiments ± s.e.m.

C   HEK293T cells were reverse transfected with pooled siRNAs targeting *GANAB* or *PRKCSH*. After 24 h, the indicated Wnt3 NanoLuciferase constructs were transfected together with a CMV Firefly luciferase reporter. 48 hours later, luciferase signals were measured in the medium and lysate. % reporter activity denotes the Wnt3 NanoLuciferase signal in the medium normalized to NanoLuciferase and Firefly luciferase signals in the lysate. Results are shown as averages of three independent experiments ± s.e.m.

D   Schematic depiction of a hypothetical mechanism where Wnt3 secretion is controlled by glucosidase II.

model for canonical Wnt signaling activation, which harbor no known mutations in the Wnt pathway. Furthermore, HEK293T cells feature an inactive state of canonical Wnt signaling, which is why the pathway can be activated by overexpression of several key components (Wnt3, Dvl3, and β-catenin).

Overexpression of Wnt3 mimics auto- and paracrine activation of canonical Wnt signaling at the level of the Wnt secreting cell which has been shown to be dependent on the Wnt-secretory components

Porcn and Evi/Wls (Bänziger *et al*, 2006; Bartscherer *et al*, 2006; Bartscherer & Boutros, 2008; Herr & Basler, 2012). In contrast, overexpression of Dvl3 induces the pathway downstream of the receptor complex in the receiving cells. Overexpression of β-catenin leads to pathway activation downstream of APC (Figs 4B and EV3A). We observed that knockdown of each of the tested candidate genes followed by pathway activation induced by Wnt3 expression resulted in strongly reduced activation of a TCF4/Wnt reporter,

which mimics transcription activation of genes regulated by β-catenin (Fig 4B). Interestingly, knockdown of *GANAB, PRKCSH,* or *UGP2* did not show a strong effect on reporter activity or even enhanced induction upon transfection with Dvl3 or β-catenin expression plasmids (Fig 4B). These results allow to conclude an interference of the candidates investigated at the level of Wnt secretion or at the receptor level, since the negative effect on Wnt activity is abolished upon further downstream pathway activation by Dvl3 or β-catenin.

To further investigate the role of the glucosidase II complex and by this protein glycosylation, secretion and quality control of glyco-protein folding in the ER in the context of Wnt signaling, we performed a Wnt secretion assay upon knockdown of *PRKCSH* and *GANAB* (Fig 4D; D'Alessio & Dahms, 2015). For this, we coupled Wnt3 to a NanoLuciferase (Hall *et al*, 2012) sequence within a Wnt3 expression plasmid. The NanoLuciferase sequence was integrated either after the signal peptide (NLucWnt3) or at the C-terminus of Wnt3 (Wnt3NLuc) to exclude an effect of NanoLuci-ferase coupling on Wnt3 secretion. A NanoLuciferase readout subsequently allowed to detect secreted Wnt3 proteins in the cell culture supernatant and to normalize it to the amount of Wnt3 in the cell lysate. Upon knockdown of either GANAB or PRKCSH, Wnt3 secretion was reduced about 40–50% using either the NLucWnt3 or Wnt3NLuc constructs (Figs 4C and EV3C). These data substantiate an already published necessity of Wnt ligand glycosyla-tion for successful secretion of Wnt proteins (Fig 4D; Komekado *et al*, 2007).

### Similarity of interaction profiles predicts functional relationships of genes

Several studies have previously shown that functionally similar genes can be identified by comparing their interaction profiles. Here, the vectors of interaction scores across query genes are compared for all possible pairs of target genes using a measure of similarity—most commonly their correlation. Two target genes with highly correlating interaction profiles are then predicted to share biological function through guilt by association (Fig 1D). Encouraged by the observation of pathway enrichment among target genes predicted to interact with the same query, we reasoned that an analysis of inter-action profile similarity should also be possible based on our results despite a relatively low number of query genes (167 after aggrega-tion of highly similar query genes). Consequently, we correlated Pearson's correlation coefficients of π-score interaction profiles for all pairwise combinations of target genes. We reasoned that data about known protein complex co-membership should be able to serve as a reference to estimate the predictive power of our approach. Hence, we downloaded all human protein complex data from the CORUM (Ruepp *et al*, 2010) database and compared our predicted associations to the known protein complex data by receiver operator characteristic (ROC) analysis. Initially, this analy-sis revealed our predictions of protein complex co-membership to be unsatisfactory. After careful inspection of the predicted relation-ships, we noticed that the correlation coefficient was in most cases influenced considerably by very small π-scores. Such data points do not hold much biological information as they merely indicate that there might be no connection between a target and a query gene based on a viability phenotype. Hence, we hypothesized that

excluding interactions with very low π-scores should shift more weight onto more informative data points and should therefore lead to more meaningful predictions of co-functionality. We conse-quently excluded all interactions with π-score < 0.2 and repeated the above analysis. As excluding interactions with a low π-score violates the Pearson's correlation's assumption of normality, we used the nonparametric Spearman's correlation instead. We calcu-lated this correlation for all pairs of target genes where at least five pairwise complete data points were available. Repeating the ROC analysis as described above revealed a considerable improvement of the resulting predictions leading to results superior to random assignment (Fig 5A). In order to identify the most suitable parameter thresholds, we systematically repeated this analysis using different combinations of the $\pi_{min}$ (minimum π-score to be consid-ered) and $n_{min}$ (minimal number of pairwise complete data points) parameters. We noticed that more conservative parameter thresh-olds lead to higher performance at predicting protein complexes. However, the more conservative these thresholds become the more genes have to be excluded from the analysis due to insufficient data. Therefore, we decided to select $\pi_{min} = 0.2$ and $n_{min} = 15$ as parame-ters for downstream analyses, assuming these cutoffs to present a good compromise between the predictive power of the analysis and the number of genes that can be considered. Based on these parame-ters, we found that our analysis holds power to correctly associate many closely interacting genes, such as *CTNNB1* and *TCF7L2,* which together form the TCF/β-catenin transcription factor complex (Morin *et al*, 1997) or the *WNT10A/FZD5* ligand receptor complex (Voloshanenko *et al*, 2017; Fig 5B). Similar interaction profiles could also be found for several members of the mediator complex, a multisubunit complex important for the transcriptional regulation of RNA polymerase II (Fig 5C).

We used a strict cutoff to select all target gene pairs for which the adjusted asymptotic *P*-value of their profile similarity (Spear-man's correlation) was smaller than 1.5e-05 and connected them to a network. The resulting network showed an edge-to-node ratio comparable to previously reported yeast networks (Costanzo *et al*, 2016) with an edge representing on average an interaction profile correlation of 0.85 (Fig EV4D). We visualized the network applying a force-directed spring-embedded layout that can position highly similar genes proximal to each other (Fig 5D). We next used spatial analysis of functional enrichment (SAFE; Baryshnikova, 2016a,b) to identify regions in the network enriched for specific biological processes as annotated by Gene Ontology (GO; Ashburner *et al*, 2000; Fig 5E). SAFE analysis revealed clustering of 19 subnetworks, which were associated with 217 different GO terms and comprised in total 2,479 genes.

In order to ensure that the observed modules do in fact resemble biologically meaningful functional clusters and are not just random artifacts of the analysis, we performed a random permutation analy-sis (Fig EV4A–C). As expected, we observed that upon random reshuffling of links while keeping the genes and edge number the same, the network loses its modular structure, resulting in one big cluster of genes in the center of the network. SAFE analysis reveals that this cluster enriches for metabolism genes, indicating that there is a general overrepresentation of metabolism genes among genes found to behave differentially in cancer cells.

Functionally enriched clusters not only cover biological processes commonly found to be implicated in cancer (e.g., "cell

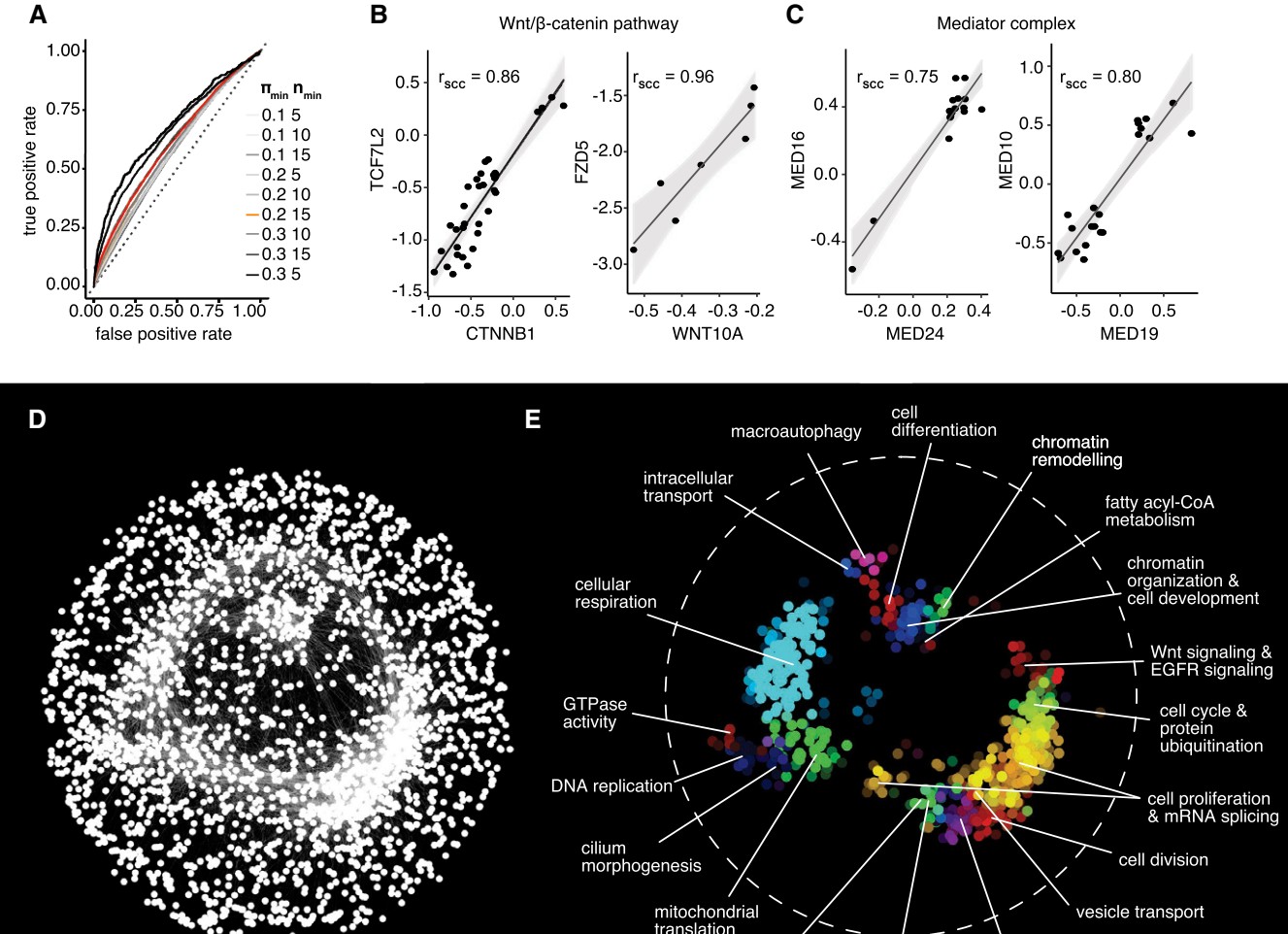

**Figure 5. Highly correlating interaction profiles can predict functional similarity.**

A    ROC curve displaying the performance of interaction profile similarity at predicting protein complex co-membership. Curves are shown for different filtering parameter combinations. The curve corresponding to the parameter combination used for downstream analysis ($\pi_{min}$ = 0.2; $n_{min}$ = 15) is highlighted in red. A gray dashed line indicates the performance expected by random assignment.

B, C    Examples of protein complexes where complex members display highly correlated interaction profiles ($r_{SCC}$ = Spearman's correlation coefficient).

D    Network of genes with highly correlated interaction profiles. In total, 2,497 nodes (genes) are connected by 19,044 links (FDR of individual links < 1.5e-05). An edge-weighted spring-embedded layout was used to position the nodes.

E    Spatial enrichment analysis with the SAFE algorithm highlights network modules consisting of genes with similar functional annotations based on Gene Ontology biological processes. The labels in the figure summarize the GO terms associated with each module.

Source data are available online for this figure.

division", "Wnt & EGFR signaling", or "cell differentiation") but also processes of general importance in cellular development and behavior (e.g., "cilium morphogenesis", "intra cellular transport", and "macro autophagy"). This implicates that the approach presented here is indeed capable of identifying novel regulators of known pathway assemblies and previously uncharacterized members of known functional biological processes. This way, we created an unprecedented resource of functional gene clusters to be exploited by future studies for deeper understanding of novel mechanisms influencing known bioprocesses, not only important in cancer but covering a wide range of biology. This resource can also

be used to validate prior assumption of gene functions in any functional study. We anticipate that as data in more cell lines and phenotypes become available this functional map of a cell will continue to grow and improve.

## Discussion

To identify novel functions of known genes or to assign cellular function to unknown genes, forward genetic screens have been conducted in many model systems ranging from bacteria to human

cells (Boutros & Ahringer, 2008). Combining high-throughput screening methods with the ability to reliably knock out every gene in the human genome by programmable nucleases now opens up the possibility of studying the consequences of complete or partial loss-of-function mutations with unprecedented accuracy in various mutational backgrounds. Genome-wide screens, predominantly for gene essentiality, have been performed and have identified a large number of known, new and context-specific essential genes (Wang *et al*, 2014, 2015; Hart *et al*, 2015; Evers *et al*, 2016; Morgens *et al*, 2016; Zhan & Boutros, 2016; Rauscher *et al*, 2017). We developed a computational approach to integrate dozens of high-throughput CRISPR/Cas9 viability screens independent of screen size, library, Cas9 type, and screening protocol. Because, compared to other techniques, CRISPR/Cas9 screens have shown to be a more sensitive method by which perturbation-induced phenotypes can be discovered in human cells (Hart *et al*, 2015; Wang *et al*, 2015), such an approach shows great promise for the systematic discovery of cancer vulnerabilities. We developed MINGLE, a computational framework that integrates CRISPR/Cas9 screens of diverse origin to map genetic interactions in cancer cells. We applied this approach to integrate data from 85 screens in human cancer cell lines and analyzed the viability effects of CRISPR/Cas9 perturbations in the context of the cell lines' genetic backgrounds. By systematically evaluating 2.1 million combinations of genes, we uncovered genetic wiring maps including many known and novel dependencies between genes implicated in tumorigenesis and resistance to therapy. We further show that these maps can identify new regulators of pathways that play important roles in specific cancer types, for example, β-catenin-dependent Wnt signaling.

Here, we demonstrate that members of the glucosidase II complex control signaling activity by regulation of Wnt3 ligand secretion, probably mediated by protein N-glycosylation. N-linked glycosylation is an ER-based process essential for protein secretion and folding (Xu & Ng, 2015; Fig 4D). Whereas N-linked glycosylation of Wnt3a has already been described in the past (Smolich *et al*, 1993), the importance of Wnt ligand glycosylation for secretion and pathway activation is controversially discussed. While some authors state a clear correlation between Wnt ligand glycosylation and secretion in a human cell line (Komekado *et al*, 2007), others could not observe loss of protein secretion upon suppressing protein N-glycosylation in *Drosophila* (Herr & Basler, 2012; Tang *et al*, 2012). Our results support a role of three genes involved in protein glycosylation on Wnt pathway activation, which could be further supported by a reduction of Wnt ligand secretion upon knockdown of *GANAB* and *PRKCSH*.

Traditionally, genetic interactions have been examined by simultaneous perturbation of two genes. Our analysis is based on the idea that one of these perturbations can be mimicked by genetic alterations that naturally occur in cancer cells. Even though we find that this concept can indeed be applied to efficiently identify true interactions, it poses a number of challenges. First of all, genetic alterations of each gene have to be pooled demanding certain assumptions about the similarity of their functional consequences. In nature, however, these assumptions do not always hold true which can confound the analysis. In this study, we have attempted to address this issue by dividing alterations into logical groups, for example, by pooling nonsense mutations and frame-shift mutations as loss-of-function variants. We have further refined these annotations by

manual curation excluding cell lines with variants known to be functionally distinct from others. Although this is currently only possible for well-characterized genes, we are confident that future advances regarding the functional characterization of cancer variants will greatly benefit our approach. It is important to point out that although absence of gene expression should be functionally similar to a complete loss of gene function due to mutation, we have not taken information about non-expressed genes into account. This is due to the fact that transcriptomic profiles of cancer cell lines have mostly been derived from microarray experiments so far. Therefore, it is challenging to distinguish between non-expressed genes and genes that are expressed at a low level (Mirnics *et al*, 2001). We believe that once RNA-seq data become broadly available for cancer cell lines, this issue can be overcome. Another challenge is posed by the fact that some genetic alterations are correlated because they co-occur in the same cell lines or cancer types. An example is the deletion of the chromosome 9p21 locus where the tumor suppressor *CDKN2A* is located. *CDKN2A* is often co-deleted with its neighboring genes (Muller *et al*, 2015), and it is thus not easily possible to understand which of them is the true driver behind a proposed interaction. This can further introduce a bias into the genetic similarity network. In our study, we address this by aggregating fully correlated query genes into "meta genes" that we then proceed to use to calculate interactions and generate the genetic similarity network. To avoid bias, we further calculate correlations of genetic interaction profiles based on only a subset of query genes such that no two query genes are more similar than 70% in terms of their cell line composition.

In this study we have required a gene to be altered in at least three different cell lines for it to be considered as a query gene for interaction analysis. As more data become available, however, we expect the number of possible query genes at this threshold to grow rapidly, which can impose a considerable multiple testing burden on our approach. Hence, we believe that this cutoff should be re-evaluated when the analysis is repeated with a larger dataset in order to find the best compromise between gene coverage and statistical power.

It has previously been demonstrated that profiles of synthetic genetic interactions can group functionally related genes through "guilt by association". Studies in human cells have formerly relied on RNA interference. However, it has been shown that this method has limitations, such as off-targeting and dosage compensation effects, that can be overcome by CRISPR/Cas9. Our approaches allowed us to analyze interaction profiles using data from many high-throughput CRISPR/Cas9 experiments. These profiles hold power to predict functional relationships of genes as we show by benchmarking against the CORUM protein complex database. Since physical protein interactions as they occur in protein complexes represent only a subset of possible functional relationships, we believe that this benchmarking can be interpreted as a lower bound for the predictive power of the analysis. We created a network that groups genes into clusters with enriched functional profiles. Findings from this analysis may be important for two reasons: First, hypotheses about the function of weakly characterized genes that are frequently deleted in cancer cells can be generated by looking at the common interaction partners within functional network modules; and second, such a network may serve as a powerful tool to infer the function of entirely uncharacterized genes based on the

function of connected genes. For example, over 10% of the genes in our network are not annotated with GO biological processes.

In its current state, a limiting factor of this type of analysis is the amount of available data. At present, there are approximately 200 genes that have been found to be frequently altered in the cell lines included in our data and for which synthetic genetic interactions can be tested. Therefore, only genes that interact with these genes can be examined currently. Nevertheless, this number will increase rapidly as new data are published, which will then allow for the creation of increasingly complex interaction networks. Pooling functionally related alterations of different genes as we demonstrate at the example of *RNF43* and *APC* can further expand the set of possible query genes. All in all, we believe that the presented approach can be a powerful way to systematically discover synthetic genetic interactions that may be of clinical interest. Furthermore, we believe that it can serve as an important asset to the quest toward more complete understanding of how human genes function. The presented workflow scales well as increasing amounts of data are becoming available.

We expect many more CRISPR/Cas9 screens in various cell lines to be carried out in the future. We will expand our analysis once these data become available to improve and diversify our findings. Finally, we aim to extend our analysis to also include data from other experiment types such as physical interactions derived from protein–protein interaction studies. Most synthetic genetic interactions, for example, do not link genes that are members of the same pathways but instead they connect members of two interacting pathways (Kelley & Ideker, 2005). Therefore, integrating synthetic interactions and physical interactions derived from protein-protein-interaction experiments might provide important new insights into how biological pathways interact with each other.

We further aim to make the predicted interactions available for browsing and download through the GenomeCRISPR database, as we believe that they can be a useful resource to inform candidate gene selection for experiments that cannot be carried out at a genome-wide scale. These include, for example, *in vivo* screens in genetically engineered mouse models that are often limited by the number of cells that can be transfected or pairwise perturbation experiments as they are now conducted in human cells using CRISPR/Cas9 (Du *et al*, 2017; Shen *et al*, 2017), which are limited, by the number of possible gene combinations.

# Materials and Methods

## Genetic profiles of cancer cell lines

To generate profiles of genetic alterations in GenomeCRISPR (Rauscher *et al*, 2017) cancer cell lines, we relied on data publicly available in the COSMIC Cell Lines Project (Forbes *et al*, 2017), the Cancer Cell Line Encyclopedia (CCLE; Barretina *et al*, 2012), and additional data published previously by Bürckstümmer *et al* (2013) for the KBM7 cell line and Klijn *et al* (2014) (Fig 1B). Taken together, these data can characterize all except for two (a patient derived glioblastoma cell line and the RPE1 cell line) cell lines currently included in GenomeCRISPR. In total, 60 different cell lines were included in the analysis. For each of these cell lines, a list of altered genes was generated, taking into consideration the following

types of alterations: (i) gain of copy number events, (ii) loss of copy number events, (iii) somatic mutations, excluding silent mutations and in-frame insertions or deletions, and (iv) mRNA overexpression.

### Selection of copy number alterations

First, copy number data were downloaded from the COSMIC Cell Lines Project v81, the CCLE (file dated 27-May-2017) and the Klijn *et al* (2014) publication. Gain and loss of copy number status was determined for each gene as follows: COSMIC provides a label for each copy number event that indicates whether the event can be classified as a gain or loss of copy number event. We adopted this classification for our analysis. In the paper by Klijn and colleagues, amplification and deletion of a gene were defined as $> 1$ or $< -0.75$ of the ploidy corrected copy number (Mermel *et al*, 2011; Klijn *et al*, 2014). Consequently, the same thresholds were used in our approach. Finally, CCLE provides $\log_2$-transformed copy number fold changes between healthy samples and cancer cell lines at the gene level. The absolute copy number of each gene per cell line was estimated from the fold change data as

$$C = [2^x \times 2]$$

where $C$ is the absolute copy number and $x$ is the $\log_2$ fold change between cell line and healthy sample. In order to assess whether this provides a realistic estimate of the total copy number, we analyzed the derived copy number for all Y chromosome genes in female cell lines where copy numbers of 0 were robustly estimated. Finally, we downloaded pre-processed gene-level copy number data from COSMIC. All genes where a copy number of 0 was estimated in a cell line were marked as loss-of-function genes. Copy number alteration events that were observed robustly across at least 2 different data sources were kept for downstream analysis after excluding alterations on the X and Y chromosomes.

### Selection of somatic mutations

Somatic mutation data were downloaded from COSMIC Cell Lines Project (version 81), the CCLE (Oncomap3 mutations dated April 10, 2012, and Hybrid Capture mutations dated May 05, 2015), and the Klijn *et al* and Bürckstümmer publications. Missense mutations and frame-shift mutations were selected, and mutations reported in disagreement between individual data sources were excluded. Next, missense mutations were classified into driver and passenger and driver as proposed by Anoosha *et al* (2016). Putative passenger mutations were excluded, and the remaining mutations were kept for downstream analysis. After pooling copy number alterations and somatic mutations, we kept all genes as query genes where an alteration was observed in at least three different GenomeCRISPR cell lines.

### Selection of overexpressed genes

In order to define genes that are overexpressed in cell lines included in GenomeCRISPR, RMA (Irizarry *et al*, 2003) normalized microarray mRNA expression data were downloaded from CCLE (CCLE_Expression_2012-09-29.res dated October 17, 2012) and the COSMIC Cell Lines Project (v81). ComBat (Leek *et al*, 2012) was

used to remove batch effects between the two different data sources, and expression levels for cell lines featured in both sources were aggregated by computing the mean. Next, gene expression *Z*-scores were computed for each gene in each cell line. Genes on the COSMIC list of cancer census genes for which a *Z*-score > 2 was observed in at least five different GenomeCRISPR cell lines were kept for downstream analysis.

## Analysis of CRISPR/Cas9 screens

To compare viability phenotypes of high-throughput CRISPR/Cas9 screens, aggregated gene-level CRISPR scores were calculated for each experiment. First, all negative selection screens for cell viability were downloaded from the GenomeCRISPR database (Rauscher *et al*, 2017). First, all genes targeted by less than three sgRNAs and all sgRNA where < 30 counts were observed in the time point 0 (T0) sample, were removed from each screen individually. In addition, we excluded all sgRNAs in the GeCKOv2 library (Sanjana *et al*, 2014) that were flagged as "isUsed = FALSE" in the "Achilles_ v3.3.8.reagent.table.txt" (https://portals.broadinstitute.org/achilles/ datasets/7/download) on the Project Achilles (Aguirre *et al*, 2016) website. After filtering, raw read counts were corrected for differences in sequencing depth by dividing the each read count by the median of all read counts of samples at both T0 and the final time point. Based on these values, fold changes were calculated for technical replicates, after adding 1 to each count to avoid logs of 0, as

$$fc_{\text{sgRNA}} = \log_2\left(\frac{rc_{\text{sample}}}{rc_{T0}}\right)$$

where $rc_{sample}$ is the normalized read count measured in the sample cell population and $rc_{T0}$ is the normalized read count measured at time point 0. In some cases, the read count abundance in the plasmid DNA pool was given instead of time point 0 sequencing data of cells. In these cases, the plasmid DNA read counts were used to calculate the fold changes for all sample replicates of those screens. Furthermore, in two cases (Doench *et al*, 2016; Munoz *et al*, 2016), no read count data were available. Here, we used the original fold change values provided by the authors of the experiments.

In order to assess the quality of each screen, Bayesian Analysis of Gene Essentiality (BAGEL; Hart & Moffat, 2016) was used to predict gene essentiality. Using precision–recall curves the ability to separate core-essential and non-essential genes based on the fold change data was examined. All screens where an area under the precision-recall-curve of less than 0.85 was observed were excluded from further analysis. After selecting screens for downstream analysis (Dataset EV4), gene-level CRISPR scores were calculated as the average fold change of all sgRNAs targeting a gene. We then used quantile normalization to normalize CRISPR scores across experiments.

## Gene-level correction of library batch effect

In order to estimate batch effects introduced by the use of different libraries, a robust linear model of the form $y_i = \beta_0 + \beta_1 x_{i1} + \ldots + \beta_n x_{in} + \varepsilon_i$ with $\beta_0 = 0$ and $y_i = y_{CRISPR,i} - Median(y_{CRISPR})$ was fitted for each gene individually where $n$ is the number of libraries including the gene, $i$ is the index of a data point, and

$y_{\text{CRISPR}}$ are quantile-normalized CRISPR scores. The coefficients $\beta_1 \ldots \beta_n$ are then the estimated difference between the CRISPR scores screened in a library to the median CRISPR scores across all libraries. A robust *F*-test as implemented in the R package "sfsmisc" was used to test the null hypothesis that the median CRISPR score observed for a gene is the same across all libraries. The Benjamini–Hochberg method (Benjamini & Hochberg, 1995) was used to estimate the false discovery rate (FDR) for each test. In case the null hypothesis could be rejected at 5% FDR, a library specific batch effect was assumed and CRISPR scores observed using that library were centered by subtracting its distance to the median of CRISPR scores across all libraries. A library was flagged from batch correction in cases where a similar (same sign of the model coefficients) batch effect was predicted for the libraries used in the screens of Wang *et al* (2017) and Tzelepis *et al* (2016). Both of these libraries were used to screen primarily acute myeloid leukemia (AML) cell lines, and thus, the null hypothesis described above might not hold true in the case of AML-specific genes. Therefore, in such cases, no batch adjustment was performed.

## Quality control of normalized CRISPR scores

To assess the appropriateness of the normalization steps described above, quality control was performed examining several different properties of the normalized data. First of all, samples were clustered to evaluate whether biologically related samples clustered more closely than more biologically distant samples. Here, the set of genes shared across all libraries was determined and Ward clustering (as implemented in R's "ward.D2" method for hierarchical clustering) was performed. The "pheatmap" R package was used to visualize the heat map shown in Fig 2A. Next, differences in normalized CRISPR scores across samples were observed at the examples of nine core-essential polymerases, and nine genes situated on the Y chromosome, all of which were sampled randomly from the set of core-essential polymerase genes (Hart *et al*, 2017) and the set of Y chromosome genes, respectively. Only screens in female cell lines were plotted in Fig 2C. To examine whether normalized CRISPR scores could distinguish core-essential genes (Hart *et al*, 2017) from non-essential genes (Hart *et al*, 2015), precision–recall curves were generated for each screen using the ROCR R/Bioconductor package (Gentleman *et al*, 2004; Sing *et al*, 2005). Further, a number of control oncogenes (KRAS, NRAS, BRAF and PIK3CA) were selected to see whether an expected difference in response to gene knockout depending on the mutation status of the gene could be observed. *P*-values shown in Fig 2E–H were calculated using a two-sided Student's *t*-test as implemented in R. Finally we checked that potential unwanted effects introduced by the batch correction did not distort findings published in the papers where data were included in our pipeline. For these comparisons, normalized CRISPR scores were used for the cell lines featured in the original publications.

## Combinatorial testing of gene–gene interactions

To test for differences in fitness response based on loss-of-function genotypes, fitness scores for all CRISPR/Cas9 screens in cell lines where genotype information was available were selected. We selected all genes that were marked as altered by somatic mutations

or copy number changes in at least three or marked as overexpressed in at least five distinct cell lines as query genes. In total, 221 genes were selected. Consequently, we identified all combinations between these query genes and genes perturbed in screens (target genes). Target genes were selected such that fitness scores were available for at least three distinct cell lines with and without a query loss-of-function. Overall, we identified ~3.8 million such combinations. As input data for the test, we used normalized CRISPR scores as described above. We fitted a linear mixed-effects model for each combination, modeling the loss-of-function genotype as fixed effect and the cell line as random effect to account for cell line-specific biases. For modeling, the R package "lme4" (Bates *et al*, 2014) was used. The R package "lmerTest" (Kuznetsova *et al*, 2016) was used to calculate an estimation of significance (*P*-value) for the coefficients of each model. After testing, similar queries were identified by calculating the Pearson's correlation of the estimated model coefficients for each pair of query genes. Pairs of query genes with a 100% correlation were merged together into a "meta" query gene. To control the expected fraction of false discoveries made during multiple testing, independent hypothesis testing (IHW; Ignatiadis *et al*, 2016) was used using the variance of the normalized CRIPSR scores of the altered (mutated or overexpressed) group as a covariate for hypothesis weighting (Fig EV2C and D).

### Quantification of genetic interactions

Interactions between genes were quantified using the $\pi$-score statistic (Horn *et al*, 2011; Laufer *et al*, 2013; Fischer *et al*, 2015). $\pi$-scores were calculated using the "HD2013SGImaineffects" function implemented in the R/Bioconductor package "HD2013SGI" (Laufer *et al*, 2013). To generate the input for the "HD2013SGImaineffects" function, normalized CRISPR scores were entered by subtracting column means and scaled by dividing columns by their standard deviation.

### Gene set enrichment network

To generate the gene set enrichment network shown in Fig 3F, we selected 10 query genes and all target genes interacting with these queries at FDR < 20%. The resulting list of edges was visualized in Cytoscape (Shannon *et al*, 2003) using a force-directed spring-embedded network algorithm. Query gene nodes were arranged manually. ConsensusPathDB (Kamburov *et al*, 2013) was used to perform gene set overrepresentation analysis, and for each query gene, a pathway term was selected from the list of results. The q-values displayed in Fig 3F are as provided by ConsensusPathDB. We would like to mention that Fig 3F was inspired by a previous study by M. Brockmann and colleagues (Brockmann *et al*, 2017).

### TCF4/Wnt-luciferase reporter assay

HEK293T cells were cultured in Dulbecco's MEM (GIBCO) supplemented with 10% fetal bovine serum (Biochrom GmbH, Berlin, Germany) without antibiotics. Experiments were performed in a 384-well format using white, flat-bottom polystyrene plates (Greiner, Mannheim, Germany). HEK293T cells were reverse transfected with 20 nM indicated siRNAs with the help of 1% of Lipofectamine RNAiMAX Transfection Reagent (#13778150; Thermo Fisher

Scientific Waltham, MA, USA). 24 hours later, cells were transfected with 0.2% of TransIT-LT1 transfection reagent (731-0029; Mirus/VWR, Madison, USA), 20 ng of TCF4/Wnt Firefly luciferase reporter (Demir *et al*, 2013), and 10 ng of actin-*Renilla* luciferase reporter (Nickles *et al*, 2012), and the canonical Wnt signaling was induced by addition of the Wnt3(20 ng)-, β-catenin (20 ng)-, or Dvl3 (5 ng)-expressing plasmids or left without induction by addition of the Ctrl plasmid pcDNA3. Luminescence was measured with the Mithras LB940 plate reader (Berthold Technologies, Bad Wildbad, Germany). TCF4/Wnt-luciferase signal was normalized to the actin-*Renilla* luciferase reporter signal. All siRNA sequences and constructs used for the TCF4/Wnt-luciferase reporter assay are listed in Dataset EV5.

### NanoLuciferase Wnt3 secretion assay

Similar to the TCF4/Wnt-luciferase reporter assay, HEK293T cells were reverse transfected with indicated siRNAs and seeded into 384-well format white, flat-bottom polystyrene plates (Greiner, Mannheim, Germany). 24 hours later, cells were transfected with 20 ng of NLucWnt3 or Wnt3NLuc expression constructs, together with 5 ng of CMV Firefly luciferase reporter plasmids (Campeau *et al*, 2009). The construct NLucWnt3 was generated by cloning the Nano-Luciferase sequence (Hall *et al*, 2012) after the signal peptide of Wnt3 into the pcDNA Wnt3 expression plasmid (Najdi *et al*, 2012), while it was cloned at the C-terminus of Wnt3 for the construct Wnt3NLuc. 48 hours later, the plates were centrifuged and 20 μl of culture medium was transferred to a new plate. NanoLuciferase signal in the lysate and medium was detected with the help of a Nano-Glo Luciferase Assay (#N1110) from Promega (USA) according to the manufacturer's instructions. Luminescence was measured with the Mithras LB940 plate reader (Berthold Technologies, Bad Wildbad, Germany). In the case of the lysate, first the signal for Firefly luciferase and then for NanoLuciferase was measured. The NanoLuciferase signal in the culture medium was normalized to the NanoLuciferase signal in lysate normalized to the Firefly luciferase signal. All siRNA sequences and constructs used for the Wnt3 secretion assay are listed in Dataset EV5.

### Gene similarity network benchmarking and modeling

In order to assess whether interaction similarity networks can predict protein complex co-membership, protein complex annotations were downloaded from the CORUM databases (Ruepp *et al*, 2010) and target genes included in the CORUM data were selected. We removed all pairwise interactions $\pi_{tq}$ with $|\pi_{tq}| < \pi_{min}$ where $\pi_{tq}$ is the interaction score between target gene $t$ and query gene $q$ and $\pi_{min}$ is a chosen threshold. Subsequently, the Spearman's correlation was calculated as implemented in the "Hmisc" R package for each possible pair of target genes using pairwise complete observations. Target gene pairs where less than $n_{min}$ data points were used to calculate the correlation were excluded. This analysis was performed for six different combinations of the parameters $\pi_{min}$ and $n_{min}$, and ROC curves were drawn to visualize how well the resulting correlations could predict protein complex co-membership as annotated in CORUM. Based on these results, and $\pi_{min} = 0.2$ and $n_{min} = 15$ were selected as thresholds to calculate Spearman's correlations between all possible target gene pairs as described above. To

take into account that each correlation is based on a different number of data points, gene pairs were ranked by *P*-value instead of raw Spearman's correlations. Hence, for each correlation, the asymptotic *P*-value was computed using the "Hmisc" R package testing the null hypothesis that the correlation between a pair of genes is 0. To select gene pairs as edges for the gene similarity network shown in Fig 5D, the false discovery rate (FDR) was controlled using the Benjamini–Hochberg method at the strict threshold of FDR < 1.5e-05. The network was visualized using Cytoscape (Shannon *et al*, 2003). A force-directed spring-embedded layout was used to position the nodes of the network without edge weighting. The visual representation of the network was inspired by previous studies in yeast (Costanzo *et al*, 2010, 2016). The spatial analysis of functional enrichment (SAFE; Baryshnikova, 2016a,b) Cytoscape plugin was used to identify functional modules in the network. For SAFE analysis, the map-based distance metric was chosen with a maximum distance threshold of 0.6 (percentile). To build the composite map, a minimal landscape size of 7 was chosen and the Jaccard distance was used as a similarity metric for group attributes with a similarity threshold of 0.75. As background for the enrichment, all nodes in the annotation standard were chosen. In SAFE, the annotation standard is a binary matrix of genes (rows) and annotation terms (columns). A value of 1 indicates that a gene is annotated with a specific annotation term. For our analysis, we generated such an annotation standard containing Gene Ontology (GO; Ashburner *et al*, 2000) Biological process annotations for all target genes tested. GO annotations were downloaded from the example data section of the SAFE algorithm's GitHub page (https://github.com/baryshnikova-lab/safe-data/blob/master/attributes/go_Hs_P_160509.txt.gz; accessed 09/13/2017) and filtered to contain only genes tested in our interaction analysis.

**Data and software availability**

Documented computer code to reproduce the analyses described in this study can be downloaded as an R package from GitHub at https://github.com/boutroslab/Supplemental-Material/tree/master/Rauscher_2017.

**Expanded View** for this article is available online.

## Acknowledgements

We thank Niklas Rindtorff, Tianzuo Zhan, Johannes Betge, and Christian Scheeder for critical comments on the manuscript. We further thank Kathrin Glaeser for sharing Wnt signaling schematics and the DKFZ Biostatistics core facility for support. Finally, we would like to thank the Boutros laboratory for helpful discussions and feedback. B.R. was supported by the BMBF-funded Heidelberg Center for Human Bioinformatics (HD-HuB) within the German Network for Bioinformatics Infrastructure (de.NBI) (Grant #031A537A). Work in the Boutros laboratory is supported in part by an ERC Advanced Grant.

## Author contributions

BR, FH, and MB designed the study. BR wrote the analysis code. TH consulted on statistical analysis. LH and OV designed and performed the experiments. All authors discussed and analyzed results. BR, FH, LH, OV, and MB wrote the manuscript. All authors read and approved the final manuscript.

## Conflict of interest

The authors declare that they have no conflict of interest.

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
