## [Review Process File · Molecular Systems Biology]

Towards an Integrated Map of Genetic Interactions in Cancer Cells

Benedikt Rauscher, Florian Heigwer, Luisa Henkel, Thomas Hielscher, Oksana Voloshanenko and Michael Boutros

Review timeline:

Submission date:	26 March 2017
Editorial Decision:	9 May 2017
Appeal received:	20 October 2017
Editorial Decision:	22 December 2017
Revision received:	5 January 2018
Editorial Decision:	19 January 2018
Revision received:	20 January 2018
Accepted:	23 January 2018

Editor: Thomas Lemberger

Transaction Report:

1st Editorial Decision

9 May 2017

Thank you again for submitting your work to Molecular Systems Biology. We have now heard back from the three referees whom we asked to evaluate your manuscript. As you will see from the reports below, the referees raise substantial concerns on your work, which, I am afraid to say, preclude its publication.

The three reviewers acknowledge that the approach taken in this study is timely and interesting. They raise however a number of serious concerns with regard to the conclusiveness of the analysis. In absence of systematic experimental benchmarking it seems therefore that the study remains, as it stands, too preliminary. As such, I am afraid that the three reviewers all rated the validity of the conclusions as "low".

In view of the recommendations provided by the three reviewers, I see no choice but to return the study with the message that we cannot offer to publish it.

Nevertheless, in view of the interest of the reviewers for your approach, we would not be opposed to consider an extended version of this study that would also include a rigorous experimental benchmarking of the results.

This would have a new number and receipt date. We recognise that this may involve further experimentation and analysis, and we can give no guarantee about its eventual acceptability. However, if you do decide to follow this course then it would be helpful to enclose with your re-submission an account of how the work has been altered in response to the points raised in the present review.

I am sorry that the review of your work did not result in a more favourable outcome on this occasion, but I hope that you will not be discouraged from sending your work to Molecular Systems Biology in the future.

REVIEWER REPORTS

Reviewer #1:

Summary

Rauscher et al generate a large, experimentally derived map of genetic interactions in cancer cells by integrative analysis of a number of CRISPR knockout screens in 55 cell lines. They use an analytical pipeline that yields a Bayes Factor for each gene in each screen, which they use as a proxy for relative fitness effect. After normalizing and controlling for batch effects using ComBat, they then compare the resulting fitness scores to cell-line specific loss of function events (mutations or copy number losses) summarized from CCLE and COSMIC. Where a given mutation is present and absent in at least three cell lines each, a genetic interaction score is calculated by comparing the essentiality scores in the wild type and mutant backgrounds.

This study describes a novel and inventive framework for learning pairwise (gene-gene) human genetic interactions from CRISPR screens in cancer cell lines. As such it represents a very promising approach for the integrative analysis of what may ultimately turn out to be a very large set of CRISPR cell line screens. However, there are a couple of critical weaknesses that must be addressed before this study can be published.

Major issues.

1. The use of Bayes Factors (output from BAGEL) as a fitness score is potentially very useful but it is not self evident that the BF is a proxy for fitness. As a binary classifier of essential/fitness genes it makes sense, but nowhere is it demonstrated that either the BF or the batch-corrected, quantile normalized fitness score is in fact a quantitative fitness score. The BF is (according to Hart & Moffat 2016) the sum of terms over all sgRNA targeting a gene, so comparing BFs across screens would likely only be relevant if they had the same number of sgRNA in each screen - unlikely given that the screens chosen for this analysis were derived from several labs each using different pooled sgRNA libraries. The authors make an effort to address this by using ComBat to remove batch effects (Fig S2F,G) but it is not at all clear that ComBat is an appropriate tool for removing batch effects from fundamentally different assays; nor is it clear that gleaning tissue specific essential genes from the top 500 most variable essentials (Fig S2F,G) necessarily translates into batch effect removal for genes with more subtle genotype-specific fitness differences. This is particularly glaring given the fact that the authors don't even use this batch-normalized data for calculating GI scores. Instead, a linear mixed-effects model is applied, using pubmed ID as an effect, but no evidence is shown that this effectively controls batch effects.

In addition to a rigorous demonstration of batch effect control, a suggestion would be to show variation in fitness scores in known backgrounds (e.g. KRAS scores in KRAS mutant vs wt; same with BRAF; CTNNB1 in APC mutants). Similar to Figure 4B-E, but with controls instead of discoveries.

2. The identification of 169 'query genes' from CCLE and COSMIC data is inspired, but the (necessary) inclusion of genomic deletions introduces some obvious artifacts that the authors fail to account for. For example, in Figure 4, the top synthetic lethal pairs in both effect size (difference in mean fitness) and significance (p-value) are GATA1 knockouts in three query backgrounds: IZUMO3, ELAVL2, and FOCAD. It is not a coincidence that all three of these genes are at chromosomal locus 9p21.3, where commonly deleted tumor suppressors CDKN2A and CDKN2B are also located. The message from the authors is that GATA1 is synthetic lethal with several genes in this locus, when a more parsimonious (and likely closer to true) explanation is that GATA1 is synthetic lethal with only one gene at this locus (e.g. CDKN2A/B) but that other genes at this locus

are frequently caught up as "passenger deletions" of these tumor suppressors and any observed digenic effect is an artifact of co-deletion. (It would indeed be news if IZUMO3, involved in sperm-egg fusion, is a cancer synthetic lethal gene).

This is a common occurrence in the list of genetic interactions in Table S4. Of the ~2300 interactions, more than 10% are either with co-located IFNA genes (secreted proteins are undetectable in pooled library screens) or olfactory receptors (unexpressed genes are also pretty unlikely to be essential in any context). The authors should find some way to account for this effect, either by collapsing query genes into a query 'locus' or by identifying the likely tumor suppressor/causal gene within these clusters.

3. While this artifact is evident from the data in the genetic interactions, it is not directly evident in the correlation network (Figure 6). However, if these correlations were measured using GI scores across correlated queries (the problem in point 2 above), the artifacts will be carried over into this network as well.

4. Published, validated examples of synthetic lethal genes are not observed or mentioned. ENO1/ENO2 (PMID 22895339), P53/POLR2A (25901683) ME2/ME3 (28099419), PARP/the entire HR pathway have been examined deeply; several of these should appear in your data, or you should explain their absence.

Minor issues.

Figure 2: What is the purpose of Fig2B? The x-axis implies all the cell lines are analyzed, but this paper focuses only on the 50 or so lines with corresponding CRISPR screens. Also 2D should be C, in both the figure and the text.

Fig 3D: haem/lymphoid are highlighted in the text CFBF, CCND3, MYB; this lineage also seems skewed for other genes (FANCD2, STK11, FTAP4, TMEM189). Possibly a batch effect? These screens are predominantly Wang et al, correct?

Fig 3E: SOS1/GRB2 are also associated with EGFR signaling (common adapter molecules for RTK signaling).

Conclusion

This is a creative approach that will ultimately yield an accurate and useful map of genetic interactions in human cancer cells, some of which might prove clinically actionable. However, the biological complexity of copy number aberrations, when viewed through the lens of digenic interactions, lead to conclusions that are extremely unlikely to be true (e.g. GATA1-IZUMO3 interaction). In the absence of experimental validation (the reviewer recognizes that this is a computational paper), we must take care to be conservative with our predictions, lest our exuberance undermine the entire field.

Even this reviewer's suggested changes might prove inadequate. The bottom line is that the authors have treated each LOF mutation in a cell line as a candidate independent query gene, when the reality is that each cell line is the sum of its mutations and thus provides its own unique genetic context. It may take far more data to effectively realize the authors' goal of a gene-gene genetic interaction network in cancer cells...but this is a good start.

Reviewer #2:

This manuscript describes a method that aims to identify genetic interactions using computational analysis of CRISPR/Cas9 screens in different human cancer cell lines. As cancer cell lines harbor relatively large number of potential loss of function mutations, and CRISPR/Cas9 induced loss of different genes affect viability in different cell lines, it is in principle possible to predict interactions even based on single-gene CRISPR/Cas9 screens, provided that sufficiently large datasets exist.

The idea behind the work is interesting, but the implementation has several shortcomings that make

the work not suitable for publication at this stage.

Major problems

1. The manuscript appears hastily written and methods are not sufficiently detailed.
2. The largest differences in the cell lines are not due to mutations causing loss of function, but due to differences in expression of genes. This is not adequately controlled. The authors should call interactions based on presence and absence of expression of specific genes in the cell lines.
3. It is not clear whether the "loss of function" mutations found in the cell lines are heterozygous or homozygous. This needs to be clarified and each type of mutation needs to be separately analyzed.
4. Different tumors have different mutations, but the driver and mutations are correlated based on the tumor type. This is not properly controlled.
5. Different tumors arise via different mutational processes, and this leads to differences in rates of mutation in fragile sites such as FHIT. This, together with differences in driver mutations leads to correlation between passengers and drivers as well. The proposed "interaction" with FHIT could be due to such an internal correlation.
6. The word "genetic interaction" implies causality and specificity. In general, it is not clear how the internal correlations in the mutation patterns within the cell lines and tumor types is dealt with. As the entire method is correlative, there is no basis for inferring causality. The authors should be clear in that they provide "predicted interactions inferred using correlation", not "genetic interactions" such as those determined using experimental methods where pairs of genes are deleted on uniform genetic background.
7. No experimental validation is provided to analyze interactions using pairwise deletions. It is thus difficult to assess the false positive rate of the predictions.

Reviewer #3:

The authors present a study to identify genetic interactions and dependencies in cancer by reanalyzing genome-wide CRISPR/Cas9 screens in cancer cell lines, taking into account their genomic backgrounds. Their approach is relevant as many more CRISPR/Cas9 screens are being performed, and by taking the genomic aberrations into account it becomes possible to identify novel interactions. This is an original and timely study that re-analyzes a large amount of publicly available data (both genomic variation and genome-wide CRISPR-Cas9 screens). I think it's an interesting concept but as the paper stands it gives me an overflow of "possible interactions" and "networks". So it is definitely an original, novel, and interesting piece of data integration, but it is not so clear what we are going to do with all these predictions.

Comments:

- 1) CDKN2A is the most frequently mutated gene - why is that not TP53? Where is TP53 ranked - can this be mentioned in that paragraph? I checked in cBioPortal on the CCLE data (somatic mutations + CNV) and found that TP53 is altered in 62% of the cases, while CDKN2A is altered in 39% of the cases. I think there is something wrong with the analysis pipeline.
- 2) Related to that, this identification of copy number variation and somatic mutations in CCLE and COSMIC has been done before, e.g., in cBioPortal, in the CCLE paper, etc. The entire first Results section is finding these mutations. It is OK to briefly recapitulate, but it should be mentioned what has been done before, and even more importantly, whether these results agree with previous efforts (which apparently they don't, see comment #1).
- 3) Clustering revealed batch effect by protocols and sgRNA libraries. Why not by cell type?
- 4) p.6 says "we selected cell lines harbouring MYC mutations". What is meant by this? I assume copy number gains? Then it's confusing to say "mutations". This has to be much more explicit, are

they gain or loss of function, what type of mutation (SNV, indel, ..), what type of consequence (non-synonymous, nonsense, ..).

5) What do we do with all the predicted interactions- will there be a website where significant interactions can be found, with explained support of sgRNA and genetic background?

Minor comments:

- page 5 second line; I believe Figure S2C refers to Figure S1C
- page 5 Figure 2 references don't make sense (there is no Figure 2C)
- page 5: "We individually performed BAGEL for each screen individually "
- page 7: Figure 3D should refer to Figure 3E instead
- page 7: some references are badly formatted

1st Revision - authors' appeal

20 October 2017

Resubmission of MSB-17-7656

We would first like to thank the reviewers for providing helpful feedback to our submitted manuscript. We would also like to thank the reviewers for their interest in our approach and the editor for giving us the possibility to resubmit an improved version of our work. We believe that, by carefully taking into account the points raised by the referees, we were able to significantly improve and extend the results presented in the previous version of the manuscript.

Specifically, the following enumeration lists all changes that were done to the figures of the manuscript in response to the reviewer comments:

Figure 1: We made changes to Figure 1B indicating the additional use of gene expression data to define differentially expressed genes as cancer variants as encouraged by Reviewer #2 in comment 2. In addition, we made a change to Figure 1C indicating that follow-up experiments were performed as encouraged by the editor.

Figure 2: In response to comment 2 of Reviewer #3 we decided to remove the original Figure 2 and replace it with a new figure focusing specifically on validation and quality control of the data normalization addressing the concerns raised by Reviewer #1 in comment 1. A complementary supplementary figure details all data normalization steps. It shows examples of how batch effects can occur, how they are addressed and provides further quality control showing for example that interactions published in previous studies from which data was used for the analysis are maintained.

Figure 3: Previous figures 3 and 4 were condensed into one new Figure 3. Here, we address the concern raised by Reviewer #1 in comment 4 by demonstrating that our approach recovers known biology. We show that, in addition to unknown interactions, well-characterized dependencies are recovered providing specific examples (such as BRAF-MAP2K1 or TP53-MDM2). We further use gene set enrichment for a more unbiased validation. An additional supplemental figure provides an overview of the correlations between query genes and these are addressed in response to points 2 and 4 raised by Reviewers #1 and #2, respectively.

Figure 4: This new figure presents follow-up experiments which were performed to confirm new regulators of the Wnt/ β -catenin signaling pathway based on their predicted interactions with known key regulators of the pathway.

Figure 5: We decided to combine figures 5 and 6 of our previous manuscript into one figure. This new figure contains a benchmarking of the predicted functional similarities by comparing to known protein complex data in order to address the concern of Reviewer #1 raised in comment 3. The figure further shows a number of known relationships in order to demonstrate that known biology can be recovered by the approach. A supplementary figure shows an additional validation of the gene similarity network, which demonstrates that structure of the network collapses upon random permutation of its links.

Below, we further provide a point-by-point response to each of the reviewers' points describing in detail how each of the points has been addressed in a new version of the manuscript.

Reviewer #1:

“Rauscher et al generate a large, experimentally derived map of genetic interactions in cancer cells by integrative analysis of a number of CRISPR knockout screens in 55 cell lines. They use an analytical pipeline that yields a Bayes Factor for each gene in each screen, which they use as a proxy for relative fitness effect. After normalizing and controlling for batch effects using ComBat, they then compare the resulting fitness scores to cell-line specific loss of function events (mutations or copy number losses) summarized from CCLE and COSMIC. Where a given mutation is present and absent in at least three cell lines each, a genetic interaction score is calculated by comparing the essentiality scores in the wild type and mutant backgrounds.

This study describes a novel and inventive framework for learning pairwise (gene-gene) human genetic interactions from CRISPR screens in cancer cell lines. As such it represents a very promising approach for the integrative analysis of what may ultimately turn out to be a very large set of CRISPR cell line screens. However, there are a couple of critical weaknesses that must be addressed before this study can be published.

Major issues.

1. The use of Bayes Factors (output from BAGEL) as a fitness score is potentially very useful but it is not self evident that the BF is a proxy for fitness. As a binary classifier of essential/fitness genes it makes sense, but nowhere is it demonstrated that either the BF or the batch-corrected, quantile normalized fitness score is in fact a quantitative fitness score. The BF is (according to Hart & Moffat 2016) the sum of terms over all sgRNA targeting a gene, so comparing BFs across screens would likely only be relevant if they had the same number of sgRNA in each screen - unlikely given that the screens chosen for this analysis were derived from several labs each using different pooled sgRNA libraries. The authors make an effort to address this by using ComBat to remove batch effects (Fig S2F,G) but it is not at all clear that ComBat is an appropriate tool for removing batch effects from fundamentally different assays; nor is it clear that gleaning tissue specific essential genes from the top 500 most variable essentials (Fig S2F,G) necessarily translates into batch effect removal for genes with more subtle genotype-specific fitness differences. This is particularly glaring given the fact that the authors don't even use this batch-normalized data for calculating GI scores. Instead, a linear mixed-effects model is applied, using pubmed ID as an effect, but no evidence is shown that this effectively controls batch effects.

In addition to a rigorous demonstration of batch effect control, a suggestion would be to show variation in fitness scores in known backgrounds (e.g. KRAS scores in KRAS mutant vs wt; same with BRAF; CTNNB1 in APC mutants). Similar to Figure 4B-E, but with controls instead of discoveries.”

We thank the reviewer for this important comment. To investigate more carefully the suitability of the BAGEL score as a proxy for fitness, we systematically compared the area under time-response-curves (time vs. fold change) to the Bayes Factor determined using the sample taken at the end of the time course. Even though we did indeed observe a high correlation between these

metrics we noticed that this only hold true for loss of viability phenotypes. However, differences between increased growth phenotypes (e.g. PTEN knockout in PTEN wild-type cells) and non-phenotypes (such as the knockout of Y-chromosome gene in a female cell line) were not captured by BAGEL. We believe that this observation makes sense given that BAGEL uses core-essential and non-essential reference gene sets to separate essential from non-essential genes while growth enhancing phenotypes are not addressed by these training sets.

Consequently, we believe that BAGEL scores can in fact be used to map genetic interactions - however, without the possibility to reveal gain of viability interactions. As for our study, we deemed it important to be able to map such interactions, we decided to adapt our strategy and use a gene level CRISPR score (average fold change of all sgRNAs targeting a gene, as demonstrated previously in Wang et al., 2017) instead. We devised a new strategy for the normalization of such scores across screens explained in detail in the new manuscript. This strategy consists of two parts – a normalization of global effects such as variants in phenotype strength induced by for example the selection time of the screen, followed by a model-based gene level batch adjustment to correct for bias introduced by different CRISPR libraries due to varying efficiency of included sgRNA reagents.

We carefully validated the results of the normalization and provide evidence that the approach appropriately controls batch effects at 5 levels:

- (1) We demonstrate that based on normalized CRISPR scores cell lines of the same type but screened in different laboratories cluster together. We further observe that the screens tend to cluster by tissue or cancer type. (Figure 2A, Figure S2F)
- (2) We demonstrate the absence of obvious batch effects by plotting CRISPR scores for randomly selected core-essential genes and Y-chromosome genes across screens (Figure 2B-C; in the case of Y-chromosome genes we show female cell lines only, as a substitute for non-targeting controls).
- (3) We further show that normalized CRISPR scores achieve high performance at distinguishing core-essential (as published in Hart et al, 2017) and non-essential genes using precision-recall-curves (Figure 2D).
- (4) In addition we examine a number of control oncogene dependencies with known effect (KRAS-KRAS, NRAS-NRAS, BRAF-BRAF, PIK3CA-PIK3CA) as suggested by the reviewer (Figure 2E-H).
- (5) Finally, we demonstrate that dependencies described in original studies where data are included in our analysis (e.g. Steinhardt et al., 2017, Wang et al. 2017) are maintained throughout the batch correction procedure (Figure S2E).

“2. The identification of 169 'query genes' from CCLE and COSMIC data is inspired, but the (necessary) inclusion of genomic deletions introduces some obvious artifacts that the authors fail to account for. For example, in Figure 4, the top synthetic lethal pairs in both effect size (difference in mean fitness) and significance (p-value) are GATA1 knockouts in three query backgrounds: IZUMO3, ELAVL2, and FOCAD. It is not a coincidence that all three of these genes are at chromosomal locus 9p21.3, where commonly deleted tumor suppressors CDKN2A and CDKN2B are also located.

The message from the authors is that GATA1 is synthetic lethal with several genes in this locus, when a more parsimonious (and likely closer to true) explanation is that GATA1 is synthetic lethal with only one gene at this locus (e.g. CDKN2A/B) but that other genes at this locus are frequently caught up as "passenger deletions" of these tumor suppressors and any observed digenic effect is an artifact of co-deletion. (It would indeed be news if IZUMO3, involved in sperm-egg fusion, is a cancer synthetic lethal gene).

This is a common occurrence in the list of genetic interactions in Table S4. Of the ~2300 interactions, more than 10% are either with co-located IFNA genes (secreted proteins are undetectable in pooled library screens) or olfactory receptors (unexpressed genes are also pretty unlikely to be essential in any context). The authors should find some way to account for this effect, either by collapsing query genes into a query 'locus' or by identifying the likely tumor suppressor/causal gene within these clusters. "

The reviewer raises an important point that we agree with. In the new version of our manuscript we address this point as follows: after scanning for genetic interactions using pairwise tests we examine similarities between query genes by looking at the correlation of the linear model estimates across target genes. This revealed amongst others expected correlations between query genes that are co-located on chromosomal locus 9p21.3 as pointed out by the reviewer. We address this by aggregating query genes with identical profiles into 'meta-genes', which we then use as queries for downstream analysis.

"3. While this artifact is evident from the data in the genetic interactions, it is not directly evident in the correlation network (Figure 6). However, if these correlations were measured using GI scores across correlated queries (the problem in point 2 above), the artifacts will be carried over into this network as well. "

We agree and we believe that by addressing this issue as described above we can reduce bias in interaction profiles that would influence the correlation network. In addition, we excluded query genes from the correlation analysis such that no two query genes are made up of more than 70% by the same cell lines. To further benchmark the validity of the correlation profiles we benchmark these by comparing them to known protein complexes (Figure 5A). We find, that the interactions hold power to predict protein complex co-membership. Additionally, we validate the resulting network by random permutation of its links (Figure S6). As expected we find that random reassignment of links removes all structure from the network confirming that the profile similarities hold biological meaning.

"4. Published, validated examples of synthetic lethal genes are not observed or mentioned. ENO1/ENO2 (PMID 22895339), P53/POLR2A (25901683) ME2/ME3 (28099419), PARP/the entire HR pathway have been examined deeply; several of these should appear in your data, or you should explain their absence. "

In the revised manuscript, we put considerably more emphasis on showing that, in addition to unknown dependencies, our approach recovers many known genetic relationships. We specifically focus on this in Figure 3 where we select query genes that are well studied and show that they interact with genes involved

in the same pathway or molecular mechanism. We highlight specific pathways using TP53 stabilization, MAPK signaling and Wnt/ β -catenin signaling as examples and further provide a more global view by performing gene set enrichment for sets of genes predicted to interact with a query.

Coming back to the example interactions listed by the reviewer we would like to briefly comment on each of them as they are interesting examples demonstrating why and how different approaches to identify genetic interactions can complement each other:

1. ENO1/ENO2: This interaction is not currently be covered by our data
2. P53/POLR2A: This interaction can occur when one copy of the POLR2A gene is co-deleted with the TP53 gene making the cells more sensitive to inhibition of the RNA POLII complex by alpha-Amanitin. POLR2A is a core-essential gene (Hart 2017) that, if knocked out by CRISPR/Cas9, always leads to a lethal phenotype. Hence, this synthetic lethal interaction is most likely the cause of a hypomorph phenotype where inhibition of POLII has more severe effects when less protein is available in the cell.
3. PARP/HR pathway: We do in fact find an interaction between BRCA2 and PARP2 ($\pi = -0.7$; $p = 4.6e-4$; Supplementary Table 4). It is, however, not clear what is going on with the other PARPs, which do not show a phenotype in any of the screens. PARP inhibitors act by inhibiting the whole family of PARPs (cite) and it might be that targeting the whole family of (at least partially) redundant proteins could lead to an increased effect. Hence, it might be more appropriate to study interactions of this type in a different experimental setting such as a drug screen or a CRISPR experiment where regions of sequence homology shared by family members are targeted.
4. ME2/ME3: This interaction can also currently not be covered by our data.

We added a paragraph to the discussion where we discuss how different findings can be generated by different perturbation methods and why this makes them complementary.

“Minor

issues.

Figure 2: What is the purpose of Fig2B? The x-axis implies all the cell lines are analyzed, but this paper focuses only on the 50 or so lines with corresponding CRISPR screens. Also 2D should be C, in both the figure and the text. “

We thank the reviewer for their suggestion. As pointed by Reviewer 3 the process of identifying mutations and copy number mutations for cell lines has been described before in various other cell lines and is not one of the main points that we would like to focus on in the manuscript. Therefore, we decided to remove Figure 2 from the manuscript and limit ourselves to describing the steps taken to annotate cell line genotypes in detail in the methods section. In addition we provide analysis code (openly available on GitHub; https://github.com/boutrosfab/Supplemental-Material/tree/master/Rauscher_2017) and the results of the annotation as supplementary data to keep the analysis transparent.

“Fig 3D: haem/lymphoid are highlighted in the text CFBF, CCND3, MYB; this lineage also seems skewed for other genes (FANCD2, STK11, FTAP4, TMEM189). Possibly a batch effect? These screens are predominantly Wang et al, correct? “

Both Wang et al. as well as Tzelepis et al. have screened predominantly AML cell lines. This provides good statistical power to predict vulnerabilities that are specific to this lineage, which we believe is why a bias towards interactions related to genes important in AML can be observed.

“Fig 3E: SOS1/GRB2 are also associated with EGFR signaling (common adapter molecules for RTK signaling). “

We thank the reviewer for pointing this out. We decided to remove Figure 3E from the manuscript. Because MYC is an important transcription factor with a broad range of target genes interactions are challenging to interpret without careful experimental follow-up. Hence, in the revised manuscript, we decided to focus on more specific interactions as presented in Figures 3C-E. We provide all other results in the form of supplementary tables and aim to also include the results into the GenomeCRISPR database so interested readers can easily access them.

“Conclusion

This is a creative approach that will ultimately yield an accurate and useful map of genetic interactions in human cancer cells, some of which might prove clinically actionable. However, the biological complexity of copy number aberrations, when viewed through the lens of digenic interactions, lead to conclusions that are extremely unlikely to be true (e.g. GATA1-IZUMO3 interaction). In the absence of experimental validation (the reviewer recognizes that this is a computational paper), we must take care to be conservative with our predictions, lest our exuberance undermine the entire field.

Even this reviewer's suggested changes might prove inadequate. The bottom line is that the authors have treated each LOF mutation in a cell line as a candidate independent query gene, when the reality is that each cell line is the sum of its mutations and thus provides its own unique genetic context. It may take far more data to effectively realize the authors' goal of a gene-gene genetic interaction network in cancer cells...but this is a good start.”

We thank the reviewer for their encouragement and for acknowledging our approach as creative and useful. We agree with the reviewer that the full genetic context of each cell line influences the way these cells react to gene perturbations and that in many cases this might mask real effects. However, our results suggest that using currently available data many known dependencies can already be identified. Hence we believe that our map of interactions can be a valuable source of information that we believe will grow rapidly as even more data is published.

We would like to further point out that a key goal of our approach is to create genetic networks as described in previous studies in yeast (Costanzo 2010,

Costanzo 2016) and *Drosophila* (Horn 2011, Fischer 2015) it is not necessarily required to obtain single LOF in cell lines as long as pairwise interactions with queries contain non-redundant biological information.

We would like to thank the reviewer for the time and effort they put into reviewing this manuscript. They raised many very good and important points that we could incorporate in a new version of this manuscript, which we believe, presents a considerable improvement over its previous version. To further demonstrate the validity of our findings we performed experiments where we could successfully confirm some of our hypotheses.

Reviewer #2:

“This manuscript describes a method that aims to identify genetic interactions using computational analysis of CRISPR/Cas9 screens in different human cancer cell lines. As cancer cell lines harbor relatively large number of potential loss of function mutations, and CRISPR/Cas9 induced loss of different genes affect viability in different cell lines, it is in principle possible to predict interactions even based on single-gene CRISPR/Cas9 screens, provided that sufficiently large datasets exist.

The idea behind the work is interesting, but the implementation has several shortcomings that make the work not suitable for publication at this stage.

Major problems

1. The manuscript appears hastily written and methods are not sufficiently detailed. “

We thank the reviewer for their comment. In our revised manuscript we have aimed to carefully describe all methods. We further provide complete and annotated source code (openly available on GitHub ; https://github.com/boutrosfab/Supplemental-Material/tree/master/Rauscher_2017) to recreate data figures. We hope that this will make our analysis approach transparent and easy to follow.

“2. The largest differences in the cell lines are not due to mutations causing loss of function, but due to differences in expression of genes. This is not adequately controlled. The authors should call interactions based on presence and absence of expression of specific genes in the cell lines. “

This is clearly an important point, which we agree with very much. In the previous version of the manuscript we tried to specifically focus on double loss-of-function perturbations in the context of synthetic lethality. However, we since realized, that this can greatly limit results of the analysis and that many more interesting interactions can be identified by including further characteristics of cell lines. Therefore, in the new version of our manuscript, we included in addition gain-of-function mutations (such as activating mutations of oncogenes, e.g. BRAF V600E mutation) and expression levels of cancer census genes (as defined by COSMIC). In Figure 3D we show, for example, that predicted interactions with BRAF V600E can recover key downstream components of the MAPK signaling pathway. In a more unbiased approach (Figure 3F), we further

show for several other query genes (e.g. overexpressed BCL2 or KRAS G12/G13) that there is enrichment for interactions among processes controlled by these genes.

“3. It is not clear whether the "loss of function" mutations found in the cell lines are heterozygous or homozygous. This needs to be clarified and each type of mutation needs to be separately analyzed.”

In our analysis we do not distinguish between heterozygous and homozygous mutations. Even though we acknowledge that this is a simplified view, it is challenging to systematically disentangle homozygosity and heterozygosity in cancer cell lines. Different, but possibly functionally similar, mutations of the same gene can occur on different alleles. Also more or less than 2 copies of an allele can be present and all of these events could potentially influence the way a mutation affects the cell. However, considering all these different events individually is currently impossible due to constraints in sample size. Consequently, some simplified assumptions have to be made in order to analyze the data. Yet, we believe that meaningful results can be achieved based on these assumptions as, in many cases, they hold true. In our new manuscript we focus more on demonstrating that our results can recover many well-known biological relationships. This is specifically addressed by Figure 3 where we both highlight specific relationships and use gene set enrichment analyses as a tool to demonstrate that genes that interact with certain queries share adequate annotations of biological pathways. We also show in Figure 5 that, by correlation, predicted interactions can be used to assemble genetic networks as shown in previous studies in yeast and drosophila. We show, that the network we generate contains biologically relevant information by benchmarking against known protein complex annotations (Figure 5A) and by disrupting the network by random permutation of its links (Figure S5A). We would further like to point out that here it is not necessarily required to have interactions based on double LOF mutants to achieve meaningful networks. In drosophila studies, for example, RNA interference was used, which creates hypomorph phenotypes.

“4. Different tumors have different mutations, but the driver and mutations are correlated based on the tumor type. This is not properly controlled.”

We agree with the reviewer that this is the case and that it needs to be controlled. Hence, we systematically analyzed correlation between query genes by comparing their predicted interactions (more specifically, the model estimates of their interactions). As expected we find clusters of correlating query genes defined by either the tumor type of the mutated cell lines or their genomic location (for example, genes co-deleted with CDKN2A; Figure S3A). First we aggregated identical query genes into ‘meta genes’ that we use for downstream analysis (Figure S3B). Further we excluded all query genes that are made up of mutations in >70% of the same cell lines as any other query gene. We used this reduced query set for correlation analyses presented in Figure 5. These steps considerably improved the results of the correlation analysis as determined by the benchmarking analysis presented in Figure 5A where we estimate the performance of our approach by looking at predicted similarities of known

protein complex partners. We further describe the phenomenon of query gene correlations in the main text of the manuscript to make readers aware.

“5. Different tumors arise via different mutational processes, and this leads to differences in rates of mutation in fragile sites such as FHIT. This, together with differences in driver mutations leads to correlation between passengers and drivers as well. The proposed "interaction" with FHIT could be due to such an internal correlation. “

We agree that this might be the case. However, we do think that the proposed interaction between CTNNB1 and FHIT makes sense. FHIT has been shown to be a negative regulator of CTNNB1 (PMID 18077326) and deletion of FHIT occurs frequently in cells that rely on active WNT/CTNNB1 signaling. These cells harbor different mutations of the Wnt signaling pathway (e.g. CTNNB1 mutation in HCT116 cells, APC truncation in HT29 cells or RNF43 mutation in HPAFII cells) and hence the interaction with FHIT cannot simply be explained by correlation with one of these. Nevertheless, in our revised manuscript we decided to first focus on known relationships to highlight that our approach produces valid results. We show in Figure 3 at the examples of MAPK signaling, Wnt/ β -catenin signaling and TP53 stabilization that predicted interactions with well-characterized query genes can recover known relationships. We then selected a number of relationships we found interesting as they were predicted to interact with several independent query genes related to Wnt/ β -catenin signaling (Figure 4A). We hypothesized that these genes might be positive regulators of the pathway and are thus required by cells that rely on active Wnt/ β -catenin signaling for proliferation and survival. We performed follow-up experiments in an independent model system (Figure 4), taking into account correlations between relevant query genes and were able to confirm our hypothesis.

“6. The word "genetic interaction" implies causality and specificity. In general, it is not clear how the internal correlations in the mutation patterns within the cell lines and tumor types is dealt with. As the entire method is correlative, there is no basis for inferring causality. The authors should be clear in that they provide "predicted interactions inferred using correlation", not "genetic interactions" such as those determined using experimental methods where pairs of genes are deleted on uniform genetic background. “

Traditionally, analyses of genetic interaction screens have so far been based mainly on the definition by Fisher (1918), which in fact does not imply that genetic interactions are causal. Our analysis is based on the same principles and therefore we do believe that it is valid. Further, it is not necessary to study pairwise gene deletion in order to infer genetic interactions. Previous studies in *drosophila*, for example, used RNA interference to introduce perturbation, which causes hypomorph phenotypes and not a complete knockout of the gene. Nevertheless we do understand the reviewer's comments regarding the specificity of predicted interactions, which needs to be taken into account when interpreting the data. As outlined above in our response to point four we address this in the revised by first aggregating fully correlated genes (Figure S3A-B) and removing excluding highly correlated query genes when generating the genetic network. We believe, and can also show, that interactions estimated by our

approach are valuable and can provide new insights into the genetic wiring map of cancer cells when interpreted appropriately.

“7. No experimental validation is provided to analyze interactions using pairwise deletions. It is thus difficult to assess the false positive rate of the predictions. “

We realize that not being able to judge the validity of the results has been a major problem of the previous version of the manuscript. Therefore, in the new version we put a lot more emphasis on showing that our approach can recover a lot of known biology. As pointed out above we provide various examples for recovered known relationships in Figure 3 and show that genes predicted to interact with query genes enrich for processes that are controlled by the query. In addition we performed validation experiments to demonstrate that based on our predictions new gene functions can be discovered. Here, we selected genes that, similar to well characterized Wnt-pathway components such as CTNNB1 or FZD5, were predicted to negatively interact with several independent query genes related to Wnt/ β -catenin signaling. We hypothesized that these genes might also be positive regulators of the Wnt signaling and that therefore cell lines with aberrant Wnt signaling depend on these genes more than others. We performed Wnt reporter assays in an independent model system (HEK293T human embryonic kidney cells). Here we activated Wnt signaling at different levels by expressing WNT3, DVL3 or CTNNB1 and observed how perturbation of candidate genes affected the activity of a WNT/TCF4 reporter. Among siRNA knockdown of target genes we observed reduced reporter activity confirming our hypothesis and showing how our approach can help to find new regulators of signaling pathways.

Reviewer #3:

“The authors present a study to identify genetic interactions and dependencies in cancer by reanalyzing genome-wide CRISPR/Cas9 screens in cancer cell lines, taking into account their genomic backgrounds. Their approach is relevant as many more CRISPR/Cas9 screens are being performed, and by taking the genomic aberrations into account it becomes possible to identify novel interactions. This is an original and timely study that re-analyzes a large amount of publicly available data (both genomic variation and genome-wide CRISPR-Cas9 screens). I think it's an interesting concept but as the paper stands it gives me an overflow of "possible interactions" and "networks". So it is definitely an original, novel, and interesting piece of data integration, but it is not so clear what we are going to do with all these predictions.

Comments:

1) CDKN2A is the most frequently mutated gene - why is that not TP53? Where is TP53 ranked - can this be mentioned in that paragraph? I checked in cBioPortal on the CCLE data (somatic mutations + CNV) and found that TP53 is altered in 62% of the cases, while CDKN2A is altered in 39% of the cases. I think there is something wrong with the analysis pipeline. “

We thank the reviewer for raising this point. The reason TP53 was not listed as the most frequently mutated gene in our previous manuscript was due to the way mutations for cell lines were selected. Specifically, we selected mutations where

we were confident that they would lead to a loss of gene function. These include frame-shift mutations and gene deletion events. The most common TP53 alterations, however, are missense mutations. In the previous version of our manuscript we excluded these as in many cases it is not easy to judge the functional impact of such mutations. However, the example of TP53 shows that this is a major limitation of the approach that we felt needed to be improved. Therefore, we built on previous work by Anoosha and colleagues (BBA, 2016) where missense mutations listed in the COSMIC database were classified into passenger and driver mutations by the frequency at which they occur across cancer samples. We found this classification to be a useful approximation. Looking specifically at the example of TP53 this allowed us to identify with high confidence several well-known interaction partners of TP53 such as MDM2 and MDM4 (Figure 3C). This further allowed us to look not only at loss-of-function mutations but also at mutations of oncogenes such as BRAF (Figure 3D) or KRAS (Figure 3F) further adding to the increased comprehensiveness of the results presented in our new manuscript.

“2) Related to that, this identification of copy number variation and somatic mutations in CCLE and COSMIC has been done before, e.g., in cBioPortal, in the CCLE paper, etc. The entire first Results section is finding these mutations. It is OK to briefly recapitulate, but it should be mentioned what has been done before, and even more importantly, whether these results agree with previous efforts (which apparently they don't, see comment #1). “

We agree with the reviewer on this point. The main intention behind this manuscript is not to demonstrate how mutations can be identified or classified into functional groups but to show how they can be linked to perturbation response. Therefore, we decided to remove Figure 2 as it was presented in the old version of the manuscript and instead focus more on key aspects of data integration and validation of the results. We moved explanations of how genotype annotations of different types and from different databases were integrated and grouped into the materials and methods section and provide final annotations as a data table (Supplementary Table 1).

“3) Clustering revealed batch effect by protocols and sgRNA libraries. Why not by cell type? “

In Figure 2 of the new version of our manuscript we display a heat map that demonstrates that cell lines of the same type do cluster together. We show that technical batch effects, introduced for example by the sgRNA library that was used, need to be corrected else they can mask similarities between biologically related samples.

“4) p.6 says "we selected cell lines harbouring MYC mutations". What is meant by this? I assume copy number gains? Then it's confusing to say "mutations". This has to be much more explicit, are they gain or loss of function, what type of mutation (SNV, indel, ..), what type of consequence (non-synonymous, nonsense, ..). “

We agree with the comment of the reviewer. The MYC gene is most frequently altered by copy number amplification and the term ‘mutation’ can be

misleading. In addition, MYC is an important transcription factor that is expected to interact with a wide variety of genes across many pathways. Hence, we decided to not focus on MYC as a case study in the revised manuscript but to instead focus on showing more specific and more easily interpretable examples (Figure 3). For all examples of genetic dependencies shown in the new version of the manuscript we have made sure to be clear about the nature of the query gene, i.e. the type of alteration that is looked at.

“5) What do we do with all the predicted interactions- will there be a website where significant interactions can be found, with explained support of sgRNA and genetic background? “

We aim to incorporate the resulting genetic interactions as an additional functionality into the GenomeCRISPR website where users could search and download predicted interactions to perform comparative analyses. We anticipate that this would also allow us to continuously update the results as we integrate more CRISPR screening data that gets published. In addition, all results including analysis scripts will also be included in the supplemental materials.

We envision various use cases where our proposed interactions could prove to be a valuable resource in addition to what we already demonstrate in the manuscript. For instance, CRISPR screens can be conducted by injecting cancer cells into genetically engineered mouse models to identify vulnerabilities of cells with specific molecular alterations *in vivo*. In such an experiment only a limited amount of cells can be injected and hence only small CRISPR libraries can be used. Here, our proposed interactions could be used to drive the design of such screens helping to select promising genes to include into the library. Further, genetic interaction studies are now also conducted using multiplexed CRISPR/Cas9 approaches to induce double (or potentially even more) gene knockouts. These studies can only investigate a limited number of gene combinations due to limitations regarding the size of the screen. Here, again, our predictions could inform the selection of candidate genes to include.

*“Minor comments:
- page 5 second line; I believe Figure S2C refers to Figure S1C
- page 5 Figure 2 references don't make sense (there is no Figure 2C)
- page 5: "We individually performed BAGEL for each screen individually "
- page 7: Figure 3D should refer to Figure 3E instead
- page 7: some references are badly formatted“*

We thank the reviewer for pointing out these mistakes and have corrected them in a revised version of the manuscript.

Thank you again for submitting your work to Molecular Systems Biology. We have now heard back from the two referees who accepted to evaluate the revised study. The reviewers are now supportive and I am pleased to inform you that we will be able to accept your manuscript for publication in Molecular Systems Biology pending only minor modifications:

- the reviewers raise some final minor points that should be addressed with suitable amendments in the text and discussion.
- for the HTML version of your paper, please provide the following items:
 - three to four 'bullet points' highlighting the main findings of your study
 - a short 'blurb' text summarizing in two sentences the study (max. 250 characters)
 - a 'thumbnail image' (width=211 x height=157 pixels, Illustrator, PowerPoint, OmniGraffle or jpeg format), which can be used as 'visual title' for the synopsis section of your paper.
 - Please complete the CHECKLIST available at http://embopress.org/sites/default/files/Resources/EP_Author_Checklist_Master.xlsx. Please note that the Author Checklist will be published alongside the paper as part of the transparent process <http://msb.embopress.org/authorguide#transparentprocess>.
 - Please rename Figure S1-S4 into Figure EV1-EV4 and place their legends in the main word file. They will be typeset and published online as expandable/collapsible 'Expanded View Figures'.
 - Please rename Table S into Dataset EV and include call out in the main text.
 - if the graph dataset is linked to a specific figure, we would suggest to rename it "Source Data for Figure xx" in which case the file can be downloaded directly from the figure. Since it is a Cytoscape file, please zip it such that the .cys extension is 'protected'.
 - please rename "source code" "Computer code EV1" and include an explicit callout in the text.

 REVIEWER REPORTS

Reviewer #1:

Summary

Rauscher et al have substantially improved this examination of genetic interactions that arise in CRISPR screens of cancer cells. The pipeline seems robust and is well explained, the emergence of expected interactions in selected backgrounds lends confidence to the approach (Fig 3), and experimental testing of predicted genetic interactions validates novel predictions. More fundamentally, the approach is tremendously useful: conceptually, this is the first large-scale effort to identify specific genetic interactions from the combination of known cell line mutations and the differential phenotype of CRISPR knockouts. Technically, it describes a method to reduce batch effects across CRISPR screens to enable pooled analyses, increasing statistical power to detect interactions. Kudos to the team for providing the R scripts and cytoscape files to view/replicate the figures and processed data.

The paper is cleanly laid out and well written. I have only a few minor points to recommend:

1. P14, combining interactions into a network - taking all gene pairs with Spearman P-value < 0.5? Is this a typo? If not, this seems somewhat lax! Perhaps a supplemental figure/figure panel showing the coverage-vs-stringency tradeoff is warranted. If it's just a typo, 0.05 is fine.
2. p15, "x different GO terms and comprised in total y genes." Would be nice to define x and y here.
3. Robustness: the method identifies RNF43-related GIs that are then experimentally validated. The data set includes the Steinhart et al study that explicitly screens RNF43 mutant cells. So the interactions are detected by their presence in 3 of 60 cell lines (5%). Would they still be detected in 3 of 400 cell lines, now that the Avana data is available? Or is the "5% rule" a reasonable lower bound for detecting interactions in large-scale data? If the authors have thoughts on this topic, it

might be worth including them in the discussion.

Reviewer #2:

The authors have addressed almost all of my concerns, and the manuscript is improved quite dramatically. I would support publication provided that the authors more directly address my original point regarding gene expression. At least it needs to be stated that a major difference between cell types is difference of expression of genes, and that the fact that a gene is not expressed has the same effect as a complete loss of function. Why this is not analyzed in the paper is unclear to me, but at a minimum it should be clearly stated in the discussion segment.

2nd Revision - authors' response

5 January 2018

Response to the reviewers of the manuscript MSB-17-7656R-Q

We thank the reviewers for their helpful comments. Below, we provide a point-by-point reply to their comments. We have changed the corresponding paragraphs of the manuscript accordingly. Changes in the text are highlighted in red.

Reviewer #1:

“Rauscher et al have substantially improved this examination of genetic interactions that arise in CRISPR screens of cancer cells. The pipeline seems robust and is well explained, the emergence of expected interactions in selected backgrounds lends confidence to the approach (Fig 3), and experimental testing of predicted genetic interactions validates novel predictions. More fundamentally, the approach is tremendously useful: conceptually, this is the first large-scale effort to identify specific genetic interactions from the combination of known cell line mutations and the differential phenotype of CRISPR knockouts. Technically, it describes a method to reduce batch effects across CRISPR screens to enable pooled analyses, increasing statistical power to detect interactions. Kudos to the team for providing the R scripts and cytoscape files to view/replicate the figures and processed data.

The paper is cleanly laid out and well written. I have only a few minor points to recommend.”

We thank the reviewer for their positive feedback.

“1. P14, combining interactions into a network - taking all gene pairs with Spearman P -value < 0.5 ? Is this a typo? If not, this seems somewhat lax! Perhaps a supplemental figure/figure panel showing the coverage-vs-stringency tradeoff is warranted. If it's just a typo, 0.05 is fine.”

This cut-off chosen is indeed 0.5. We would like to point out, however, that these P-values are corrected for multiple testing using the Bonferroni method – a very conservative method for multiple testing correction. In fact on our dataset the Bonferroni-corrected P-value of 0.5 corresponds to an FDR (as estimated by the Benjamini-Hochberg method) of 1.5e-05 and is therefore very strict.

“2. p15, “x different GO terms and comprised in total y genes.” Would be nice to define x and y here.”

We thank the reviewer for pointing this out. We have replaced the place holders with the actual values.

“3. Robustness: the method identifies RNF43-related GIs that are then experimentally validated. The data set includes the Steinhart et al study that explicitly screens RNF43 mutant cells. So the interactions are detected by their presence in 3 of 60 cell lines (5%). Would they still be detected in 3 of 400 cell lines, now that the Avana data is available? Or is the “5% rule” a reasonable lower bound for detecting interactions in large-scale data? If the authors have thoughts on this topic, it might be worth including them in the discussion.”

In fact, the dataset comprises a total of 6 RNF43 mutated cell lines. These include, in addition to the pancreatic cell lines screened by Steinhart et al., three colorectal cancer cell lines – a cancer type where RNF43 mutations are also frequently observed. Nevertheless, we agree that the minimum number of mutated cell lines required for the analysis of a variant is an important cutoff to discuss. We would argue that having a variant present in three different cell lines is a minimal requirement for the variant to be suitable for interaction analysis. With the new Avana data, however, the number of variants available for analysis increases drastically which also imposes a considerable multiple testing burden. Hence, we would most likely increase the minimum number of cell lines when analyzing a larger dataset. The ‘5% rule’ sounds like a reasonable starting point to be refined by careful comparison of different cutoffs and examination of known interactions. We have added a paragraph on this matter to the discussion of the manuscript.

Reviewer #2:

“The authors have addressed almost all of my concerns, and the manuscript is improved quite dramatically. I would support publication provided that the authors more directly address my original point regarding gene expression. At least it needs to be stated that a major difference between cell types is difference of expression of genes, and that the fact that a gene is not expressed has the same effect as a complete loss of function. Why this is not analyzed in the paper is unclear to me, but at a minimum it should be clearly stated in the discussion segment.”

We thank the reviewer for their positive feedback. We do agree that it is important to look at gene expression in the context of our analysis. In our paper, we have addressed differences in gene expression by considering gene overexpression as a genomic variant for inference of genetic interactions. We have, in addition, thoroughly considered to take loss of gene expression events into account. Problematically, transcriptomic information about cancer cell lines have so far been derived from microarray experiments where it is challenging to distinguish between genes that are not expressed and genes that are expressed at a low level. This can, however, make an important difference. Nevertheless, we are confident that as RNAseq data are now starting to become available for cancer cell lines, we will be able to look at gene expression, and specifically loss of gene expression, more specifically. As the reviewer suggests, we have added a section discussing this point to the manuscript.

Thank you again for submitting your work to Molecular Systems Biology. We are now globally satisfied with the modifications made and I am please to inform you that we will be able to accept your paper for publication pending the following minor amendments:

Main manuscript

- To clarify the issue related to the threshold used to include edges in the similarity network, it might be more intuitive to refer to the threshold used in terms of false discovery rate using the Benjamini-Hochberg method. Perhaps something along the following line: "To select gene pairs as edges for the gene similarity network, the false discovery rate (FDR) was controlled using the Benjamini-Hochberg method at the strict threshold of $FDR < 1.5e-05$ ".
- Please remove tables from the main text, label them appropriately Table nn and add them after Figures"
- Thank you very much for including suggestion for the cover. I will get back to you on this in a separate message.
- Please list 20 authors + et al instead of 1 author + et al in the reference list to match the MSB reference style. <http://msb.embopress.org/authorguide#referencesformat>

Figure legends

- Please list all the main figures first, followed by a new section with the EV figure legends.
- Please add a description of panel E in the figure legend for figure 5.

EV figures

- Please rename fig. EV2-5 to EV1-4 and update the legends and callouts accordingly (in our format, an expanded view figure does not need to be associated to a particular main figure)

EV tables

- Please rename the tabs from Supplementary Table 1-4 to Table EV1-4.

List of Supplementary Material

- This file will not be published. Please move the legends/file descriptions from 'List of Supplementary Material' to the corresponding files/legends.

Code

- We would kindly ask you to use the specific link to the code associated with the current paper (https://github.com/boutroslab/Supplemental-Material/tree/master/Rauscher_2017) rather than the more generic link to the whole repository of the Boutros lab (<https://github.com/boutroslab>). We appreciate that the README document of the repository will include the abstract and link to the MSB Paper. We would be grateful if the link back to the paper could also be added directly in the README page of the specific Rauscher_2017 folder on GitHub.

Callouts

- There is a callout to fig 1E-H on page 24, but fig 1 has only 4 panels; A-D. Should the callout be changed to fig 2E-H?
- Please make sure that the callouts to the EV datasets appear in numerical order.

The authors made the requested editorial changes and submitted the final version of their manuscript.

Corresponding Author Name: Michael Boutros
 Journal Submitted to: Molecular Systems Biology
 Manuscript Number: MSB-17-7656R-Q